



# Age of the Tibetan ice cores

Shugui Hou[1,2], Wangbin Zhang[1], Chaomin Wang[1], Shuangye Wu[3], Yetang Wang[4],

Hongxi Pang[1], Theo M. Jenk[5,6] and Margit Schwikowski[5,6]

[1]School of Geographic and Oceanographic Sciences, Nanjing University, Nanjing,

210023, China.

[2]CAS Center for Excellence in Tibetan Plateau Earth Sciences, Beijing, 100101,

China.

[3]Department of Geology, University of Dayton, Dayton, OH 45469, USA.

[4]College of Geography and Environment, Shandong Normal University, Jinan,

250358, China.

[5]Laboratory of Environmental Chemistry, Paul Scherrer Institute, CH-5232 Villigen

PSI, Switzerland.

[6]Oeschger Centre for Climate Change Research, University of Bern, Sidlerstrasse 5,

CH-3012 Bern, Switzerland.

Correspondence to: Shugui Hou (shugui@nju.edu.cn)





**Abstract.** An accurate chronology is the essential first step for a sound understanding

of ice core records, however, dating of ice cores drilled from the high elevation

glaciers is challenging and often problematic, leading to great uncertainties. The

Guliya ice core, drilled to bedrock (308.6 m in length) from the northwestern Tibetan

Plateau (TP) and widely used as a benchmark for paleoclimate research, is believed to

reach > 500 ka (thousand years) at its bottom. Meanwhile other Tibetan ice cores (i.e.,

Dasuopu and East Rongbuk in the Himalayas, Puruogangri in the central TP, and

Dunde in the northeastern TP) are mostly of the Holocene origin. In this study, we

drilled ice cores to bedrock from the Chongce ice cap ~30 km from the Guliya ice

core drilling site. We performed measurements of $^{14}$C, $^{210}$Pb, tritium and $\beta$-activity for

the ice cores, and used these values in a two-parameter flow model to establish the ice

core depth-age relationship. The modeled ages of two Chongce ice cores at the

ice-bedrock contact are $8.3\pm^{6.2}_{3.6}$ ka B.P. and $9.0\pm^{7.9}_{3.6}$ ka B.P. respectively. The

significant discrepancy between the Guliya and all other Tibetan ice core

chronologies calls for a revisit of this legend ice core record.



## 1 Introduction

Ice cores from the Tibetan Plateau (TP) provide a wealth of information for past

climatic and environmental conditions that extends beyond the instrumental period

(e.g., Thompson et al., 1989; 1997; 2000). An accurate chronology is the essential

first step for a sound understanding of such ice core records. However, ice core dating

is always a challenging task because seasonal signals suitable for annual layer

counting are usually only observable in top sections of ice cores. For deeper (older)

sections, annual cycles cannot be identified due to rapid thinning of ice layers. If

sufficient organic matter (e.g., plant or insect fragments) is found inside the ice cores,

the conventional radiocarbon ($^{14}$C) dating can be used (Thompson et al., 2002).

Unfortunately, the presence of such material is far from guaranteed, which limits its

application for ice core dating. Recently, a novel method was developed to extract

water-insoluble organic carbon (WIOC) particles at microgram level from

carbonaceous aerosol embedded in the glacier ice for Accelerator Mass Spectrometry

(AMS) $^{14}$C dating (Jenk et al., 2007; Uglietti et al., 2016). Carbonaceous aerosol is

constantly transported to the glaciers, where it is deposited and finally incorporated in



glacier ice. Consequently, carbonaceous aerosol in ice cores can provide reliable

dating at any given depth when the samples contain sufficient carbon mass (> 10 µg).

Here we applied this recently established technique for dating the Tibetan ice cores.

**2 Chronology of previous ice cores**

The Dunde ice cores are the first ones ever drilled on the TP (Fig. 1). In 1987, three

ice cores (139.8 m, 136.6 m and 138.4 m in length) were drilled to bedrock at an

altitude of 5325 m a.s.l. (above sea level) from the Dunde ice cap (38°06′ N, 96°24′ E)

in the Qilian Shan mountains on the northeastern TP (Fig. 1). Their original

chronology was suggested to be 40 ka B.P. (before present, i.e., before 1950 AD) at

the depth of 5 m above the ice-bedrock contact, and potentially more than 100 ka B.P.

at the ice-bedrock contact (Thompson et al., 1989). Later, Thompson et al. (2005)

provided a single [14]C date of 6.24±0.33 ka B.P. for a sample collected close to the

ice–bedrock contact, and suggested the possibility that the Dunde cores may have a

Holocene origin.

The Guliya ice core: In 1992, a 308.6 m ice core to bedrock was drilled at an altitude



of 6200 m a.s.l. from the Guliya ice cap (35°17′ N, 81°29′ E) on the northwestern TP

(Fig. 1). Top 266 m of the Guliya core was dated to a period spanning 110 ka B.P.,

and the ice below 290 m was suggested to be >500 ka B.P. based mainly on $^{36}$Cl-dead

ice at the bottom section (Thompson et al., 1997).

The Dasuopu ice cores: In 1997, three ice cores were drilled from the Dasuopu glacier

(28°23′ N, 85°43′ E) in the Himalayas. The first core (159.9 m in length) was drilled

at an altitude of 7000 m a.s.l., and two more cores (149.2 and 167.7 m in length,

respectively) were drilled to bedrock 100 m apart on the col at an altitude of 7200 m

a.s.l. (Thompson et al., 2000). It was suggested that the Dasuopu ice field

accumulated entirely during the Holocene (Thompson et al., 2005).

The Puruogangri ice cores: In 2000, three ice cores (118.4 m, 214.7 m and 152 m in

length) were drilled at an altitude of 6070 m a.s.l. from the Puruogangri ice cap

(33°55′ N, 89°05′ E) on the central TP (Fig. 1). The measured oldest $^{14}$C date is

6.44±0.16 ka B.P. at 210.5 m depth of the 214.7 m ice core. The dating was

extrapolated another 0.5 m further down to 7 ka B.P. (Thompson et al., 2006),

indicating its Holocene origin (Thompson et al., 2005).

The East Rongbuk ice cores: In 2001, one ice core to bedrock (117.1 m in length) was

drilled on the col of East Rongbuk Glacier (28º1′ N, 86º58′ E, 6518 m a.s.l.) on the

north slope of Qomolangma (Mount Everest) in the Himalayas. In 2002, two more ice

cores (108.8 m and 95.8 m in length, respectively) were drilled to bedrock nearby the

previously drilling site. In a previous study, we matched the $CH_4/\delta^{18}O_{atm}$ phase record

of both the East Rongbuk 117.1 m and 108.8 m cores to the GRIP $CH_4$ and the GISP2

$\delta^{18}O_{atm}$ of the Greenland summit ice cores, and the results suggest a Holocene origin

of the East Rongbuk ice cores (Hou et al., 2004).

The Grigoriev ice core is drilled at the top of the Grigoriev ice cap in the west Tien

Shan (41°59′ N, 77°55′ E; Fig. 1). In 2007, an ice core (86.87 m in length) was drilled

to bedrock at an altitude of 4563 m a.s.l.. The $^{14}C$ dating of organic soil collected from

the bottom of the ice core borehole showed that the age of the soil was 12.656-12.434

ka B.P., coincident with the beginning of the Younger Dryas cold period (Takeuchi et

al., 2014).

**3 The Chongce ice cores**

In 2012, we drilled two ice cores to bedrock with length of 133.8 m (Core 1) and

135.8 m (Core 2) and a shallow core (Core 3) of 58.8 m at an altitude of 6010 m a.s.l.

from the Chongce ice cap on the northwestern TP (35º14′ N, 81º7′ E; Fig. 1). The

direct distance between the Chongce and the Guliya ice core drilling sites is ~30 km

(Fig. S1). In 2013, two more ice cores were recovered to bedrock with the length of

216.6 m (Core 4) and 208.6 m (Core 5) at an altitude of 6100 m a.s.l. on the Chongce

ice cap (35°15′ N, 81°5′ E). The detailed positions of the five Chongce ice cores are

shown in Fig. S2. All the ice cores were transported frozen to the cold room in the

Nanjing University for further processing. The basal sediment collected from the

bottom of Core 4 was measured for the first luminescence dating, resulting in an age

of 42±4 ka B.P., which was regarded as an upper constraint for the age of the bottom

ice at the drilling site (Zhang et al., 2018).

**4 Measurements**


$^{14}$C




We performed [14]C measurements on WIOC extracted from 22 samples collected

discretely along the 216.6 m Chongce Core 4 and 9 samples along the 135.8 m

Chongce Core 2, as well as 5 samples collected from the East Rongbuk 95.8 m ice

core. The [14]C sample decontamination was performed at Paul Scherrer Institute by

removing the ~3 mm outer layer with a bandsaw in a -20 °C cold room and rinsing

with ultra-pure water in a class 100 laminar flow box. The water-insoluble organic

carbon (WIOC) fraction of carbonaceous particles in the sample was filtered onto

freshly preheated quartz fiber filters (Pallflex Tissuquartz, 2500QAO-UP), then

combusted stepwise (10 min at 340 °C; 12 min at 650 °C) using a thermal-optical

carbon analyzer (Model4L, Sunset Laboratory Inc., USA) for separating organic

carbon (OC) from elemental carbon (EC), and the resulting $CO_2$ was measured by the

Mini Carbon Dating System (MICADAS) with a gas ion source for [14]C analysis at the

University of Bern LARA laboratory. Details about sample preparation procedures

and analytical methods can be found in previous studies (Jenk et al., 2007, 2009; Sigl

et al., 2009; Uglietti et al., 2016). The overall procedural blanks were estimated using

artificial ice blocks of frozen ultra-pure water, which were treated the same way as



real ice samples. The average overall procedural blank is 1.34±0.62 µg carbon with a

$F^{14}C$ of 0.69±0.13 (Uglietti et al., 2016). Conventional $^{14}C$ ages were calibrated using

OxCal v4.2.4 software with the IntCal13 calibration curve (Bronk Ramsey and Lee,

2013; Reimer et al., 2013).

$^{210}Pb$

The accessible time range using radioactive isotope $^{210}Pb$ dating is ~150 years due to

the 22.3-year half-life of $^{210}Pb$, a product of the natural $^{238}U$ decay series. Here $^{210}Pb$

dating was performed on the Chongce 216.6 m Core 4, with a total of 52 samples

collected from the depth of 0-76.6 m. Each sample ( ~100 - 200 g) was cut parallel to

the drilling axis in a -20 °C cold room. The samples were processed according to the

standard method established by Gäggeler et al. (*1983*). The samples were melted for

24 hours after adding 0.05% (V:V) analytical reagent HCl (30%). Afterwards, 100 µL

$^{209}Po$ tracer was added to the solution to determine the yield of the separation.

Spontaneous deposition of Po on an Ag disk (15 mm diameter), which was fixed on a

wire and immersed in the liquid, was achieved during ~7 hours at 95 °C in 500 mL

Erlenmeyer flasks using a magnetic stirrer. After drying, the disks were measured by

$\alpha$-counting at the Paul Scherrer Institute. The samples were positioned in vacuum

chambers at a distance of 1mm from silicon surface barrier detectors (ORTEC,

ruggedized, 300 and 450 mm$^2$) having an $\alpha$-energy resolution of ~23 keV full width at

half-maximum at 5.3 MeV. The yield of $^{209}$Po tracer was measured via its 4.9 MeV

$\alpha$-line. Typical chemical yields were ~75%.


Tritium

Tritium measurements were performed on the Chongce 216.6 m Core 4, with 51

samples collected successively from the depth range of 6.7-11.8 m (corresponding to

a sampling resolution of ~ 0.1 m per sample), and 42 samples from the depth range of

11.8-32.0 m (corresponding to a sampling resolution of ~ 0.5 m per sample). Each

sample is ~10 g. Samples were analyzed at the Paul Scherrer Institute using liquid

scintillation counting (TriCarb 2770 SLL/BGO, Packard SA).

*β-activity*

Twenty-two samples were collected successively from top to the depth of 10.3 m of

the Chongce 58.8 m Core 3. Each sample is ~1 kg. The $\beta$-activity was measured using

Alpha-Beta Multidetector (Mini 20, Eurisys Mesures) at the National Key Laboratory

of Cryospheric Sciences, China. More details can be found in An et al. (2016).

**5 Results**

The $\beta$-activity profile of the Chongce 58.8 m Core 3 is shown in Fig. 2a. A $\beta$-activity

peak at the depth of 8.2-8.4 m was referenced as 1963 AD, while a second $\beta$-activity

peak at the depth of 4.8-5.1 m was set as 1986 AD, corresponding to the 1986

Chernobyl nuclear accident. Both $\beta$-activity peaks were also observed in the

Muztagata ice core from the eastern Pamir (Tian et al., 2007). The calculated mean

annual accumulation rate is 140 mm w.e. (water equivalent) /year for the period of

1963–2012 AD.

The tritium profile of the Chongce 216.6 m Core 4 is shown in Fig. 2b. The tritium

activity was not corrected for decay to the time of deposition, because our purpose is

to identify the apparent tritium peak (3237±89 TU) at the depth of 21.4 m, which was





attributed to the thermonuclear bomb testing during the period of 1962-63 AD. The

calculated mean annual accumulation rate is 297 mm w.e./year for the period of

1963-2013 AD.

The $^{210}$Pb activity profile of the Chongce 216.6 m Core 4 is shown in Fig. 3, which

shows an exponential decrease as a function of depth in line with the radioactive

decay law. The $^{210}$Pb activity concentrations are in the range 7.5-317 mBq/kg, but

keep relatively stable for the lower 16 samples, with an average of 11.2±2.1 mBq/kg

(not shown). This average was taken as background $^{210}$Pb (BGD) from the mineral

dust contained in the ice core and was subtracted from the measured $^{210}$Pb activity

concentrations. From the linear regression of the logarithmic $^{210}$Pb activities (BGD

subtracted) against depth (Fig. 3), the value of the axis intercept (236±33 mBq/kg)

corresponds to the $^{210}$Pb activity at the surface of the Chongce ice cap. Thus we

applied the following function to calculate the ice age, assuming a constant initial

concentration (CIC) model. We calculated 1891±15 AD at the depth of 44.09 m (i.e.

34.36 m w.e.), resulting in a mean annual net accumulation rate of 280±47 mm

w.e./year for the period of 1891-2013 AD. This value is in very good agreement with





the 297 mm w.e./year for the period of 1963-2013 AD derived from the tritium profile

of the same ice core (i.e., the Chongce 216.6 m Core 4, Fig. 2b).

$$t_s = \lambda^{-1} ln(\frac{C_0}{C_S})$$

Where, $t_s$ stands for the age of ice at a certain depth with $^{210}$Pb activities (subtracted)

$C_S$, $\lambda$ for the decay constant of $^{210}$Pb (0.03114 $a^{-1}$), and $C_0$ for the $^{210}$Pb surface

activity.

The $^{14}$C age profile of the Chongce 216.6 m Core 4 is shown in Fig. 4. We collected

the $^{14}$C samples taking into consideration of the chronology of the Guliya ice core, but

finally realized that most of the samples, especially those collected from the upper

sections, are too young to be dated with an acceptable uncertainty. For instance, we

obtained 1891±15 AD at the depth of 44.09 m from the $^{210}$Pb measurements (Fig. 3),

and the $^{14}$C ages are 0.013-0.269 ka cal B.P. at the depth of 40.11-40.97 m, and

modern to 0.430 ka cal B.P. at 50.06-50.82 m. Even though all obtained calibrated age

ranges of the uppermost four samples include the expected ages based on the $^{210}$Pb

dating results, they have large uncertainties due to the young age and the relatively

flat shape of the calibration curve in the past 500 yrs. Furthermore, anthropogenic



contribution for samples younger than 200 yrs is likely introduce an old bias in $^{14}$C

ages due to fossil fuel ($^{14}$C dead) contribution (Jenk et al., 2006). Only 2% fossil

contribution would shift the mean of the calibrated age ranges for these samples by up

to 200 yrs towards younger ages, resulting in a smaller age range close to the ages

estimated by $^{210}$Pb dating. The $^{14}$C age profile in the depth range of 80-180 m shows

large scatter and no clear increase in age (Fig. 4). This is likely caused by the

relatively young age of samples in combination with relatively large analytical

uncertainties due to the presence of high mineral dust load in the Chongce ice core.

We made use of the $^{14}$C ages (excluding the top four samples for the reasons

discussed above), the $^{210}$Pb results (Fig. 3), and the tritium horizon (Fig. 2) to

establish the depth-age relationship for the Chongce 216.6 m Core 4 (Fig. 4), by

applying a two-parameter flow model (2p model) (Bolzan, 1985 and Supplement).

The derived age estimate just above the ice–bedrock contact (10 cm w.e. above) is

$7.0\pm_{2.7}^{4.4}$ ka B.P.. At the ice-bedrock contact the estimated age is $8.3\pm_{3.6}^{6.2}$ ka B.P. Since

the model approaches infinity as the depth gets close to bedrock, this bottom age was

derived by assuming no further thinning with depth for the last 10 cm w.e.. The model



derived annual accumulation rate is 248±36 mm w.e./year and the model derived age

at the depth of the oldest $^{14}$C sample is $4.3\pm^{1.5}_{1.1}$ ka B.P., both in good agreement with

the accumulation rate of 280-300 mm w.e./year deduced from $^{210}$Pb and tritium (Figs

2 and 3) and cal. $^{14}$C age of 4.5±0.2 ka B.P.. This indicates reasonable reliability of

the model results.

For the Chongce 135.8 m Core 2, a depth-age relationship using the 2p model was

also attempted (Fig. 5). In this case the model is constrained by the $^{14}$C cal. ages and

the *β*-activity horizon of the Chongce 58.8 m Core 3 (Fig. 2), assuming a similar

depth-age relationship for the upper parts of Core 2 and Core 3, which is reasonable

given that their drilling sites are only several meters apart (Fig. S2). Although the

derived annual accumulation rate of 137±54 mm w.e./year is in good agreement with

the 140 mm w.e./year derived from the tritium horizon (Fig. 2), we find the model to

be poorly constrained for this simulation. For instance, the derived age at the depth of

the oldest $^{14}$C sample is $9.1 \pm^{7.2}_{4.0}$ ka B.P., much older than the actual $^{14}$C age (6.3±0.2

ka B.P.) at that depth. The large uncertainty (Fig. 5) further indicates that the model is

poorly constrained. Given the close proximity between the Chongce 216.6 m Core 4



and the Chongce 135.8 m Core 2 and similar bottom altitude of their drilling sites, we

used the estimated age at bedrock derived for the Chongce 216.6 m Core 4 as an

additional constraint (Fig. 6). With the additional age constraint at the bottom, the 1σ

confidence interval for the model is significantly reduced. The ice age at the bedrock

for the Chongce 135.8 m Core 2 is thus estimated to be $9.0 \pm ^{7.9}_{3.6}$ ka B.P.. This seems

to be a reasonable estimate considering 1) the so derived accumulation rate (103±34

mm w.e./year) is in relative agreement with the tritium based estimate (140 mm

w.e./year); and 2) the modeled age at the depth of the oldest [14]C sample is now

$5.2 \pm ^{1.9}_{1.4}$ ka B.P., similar to the actual [14]C age of 6.3±0.2 ka B.P. given the uncertainty

range. Although this result is far from satisfying, it is much better than the result

obtained from the model without the additional bottom age constraint.

We have noticed that the reconstructed average annual accumulation rate from the

Chongce 216.6 m Core 4 roughly doubles the rate reconstructed from the Chongce

58.8 m Core 3. It is possible that the Core 4 drilling site may receive extra snow

supplies, such as snow drifting, whereas part of the snow deposition at the Core 3

drilling site may be blown away due to wind scouring (Fisher et al., 1983). A full



understanding of this difference will require a long-term in situ observation.

Nevertheless, this illustrates that long ice cores are not necessarily older than short

ones recovered from the same region owing to local variations in accumulation rates.

## 6 Discussion

The Himalayan ice cores (Dasuopu and East Rongbuk) were previously suggested to

be of Holocene origin (Thompson et al., 2005; Hou et al., 2004). The oldest cal. [14]C

age for a sample collected down to the ice-bedrock contact of the East Rongbuk 95.8

m ice core, is 6.72±0.43 ka B.P., confirming its Holocene origin. The ice cores from

Puruogangri in the central TP, and, to a less degree, Dunde in the northeastern TP are

of Holocene origin, too (Thompson et al., 2005, Fig. 1). For the Chongce ice cores,

our estimated ages at the ice-bedrock contact ($8.3\pm^{6.2}_{3.6}$ ka B.P. for the Chongce 216.6

m Core 4 and $9.0\pm^{7.9}_{3.6}$ ka B.P. for the Chongce 135.8 m Core 2 respectively) are

either of Holocene origin, or possible origin of late deglaciation period, similar to the

result of the Grigoriev ice core in the west Tien Shan (Takeuchi et al., 2014, Fig. 1).

In both cases, the results confirm the upper constraint of 42±4 ka B.P. derived from




the luminescence age of the basal sediment sample collected from the bottom of the

Chongce 216.6 m Core 4 (Zhang et al., 2018).

It is apparent that the temporal scale of the Guliya ice core is at least an order of

magnitude older than other TP and the Tien Shan cores. Thompson et al. (2005)

considered this as evidence that the growth (glaciation) and decay (deglaciation) of

large ice fields in the lower latitudes are often asynchronous. Our new understanding

of the chronology of the Chongce ice cores that were drilled only ~30 km away from

the Guliya ice core drilling site cannot back up this evidence. Though the validity of

the Guliya chronology has been assumed repeatedly since its publication (Thompson

et al., 2005, 2017), Cheng et al. (2012) argued that the Guliya ice core chronology

should be shortened by a factor of two in order to reconcile the difference in the $\delta^{18}O$

variations between the Guliya ice core and the Kesang stalagmite records (Fig. 1 and

Supplement). It is worth pointing out that the $^{36}Cl$-dead in the bottom section of the

Guliya core provided the key evidence for the existence of ice older than 500 ka

(Thompson et al., 1997). Since $^{36}Cl$ is presumably present as water-soluble ion, it can

be easily removed from the snow or firn layer by meltwater percolation. Thus, the

absence of $^{36}$Cl could have resulted from its leaching with glacier meltwater, as

indicated by the abnormally low chloride concentrations in the bottom section of the

Guliya record (Thompson et al., 1997), although a thorough explanation is beyond the

scope of the current work. Nevertheless, much work is necessary for conviction of the

exceptional length of the Guliya ice core.

## 7 Conclusions

We provided estimation of ages at the ice-bedrock contact of the Chongce ice cores

drilled from the northwestern Tibetan Plateau, where exceptional length of the ice core

record was previously suggested. Our results suggest that the temproal scale of the

Chongce ice cores is at least one order of magnitude younger than the nearby Guliya

ice core, but similar to all the other Tibetan ice cores, confirming the recent

conclusion derived from the luminescence age of the Chongce ice core. The current

work has wide implications, such as reexamination of the Tibetan ice core records,

whether or not existance of asynchronous glaciation on the Tibetan Plateau.



Undoubtly, more reliable chronologies of the Tibetan ice cores to bedrock are

necessary and urgent, thanks to the new development of dating techniques.

**Data availability.** All data used in this paper will be deposited in a public data

archive or available upon request to the corresponding author.

**Author contribution.** SH conceived this study, drilled the Chongce ice cores, and

wrote the paper. CW, TMJ and MS measured the $^{14}$C, $^{210}$Pb, $\beta$-activity and tritium. All

authors contributed to discussion of the results.

**Competing interests.** The authors declare that they have no conflict of interest.

**Acknowledgments.** Thanks are due to many scientists, technicians, graduate students

and porters, especially to Yongliang Zhang, Hao Xu and Yaping Liu, for their great

efforts in the high elevations, to Chiara Uglietti and Heinz Walter Gäggeler for help to

measure the $^{14}$C and the $^{210}$Pb samples, and to Guocai Zhu for providing the ground



penetrating radar results. This work was supported by the National Natural Science

Foundation of China (41330526).

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





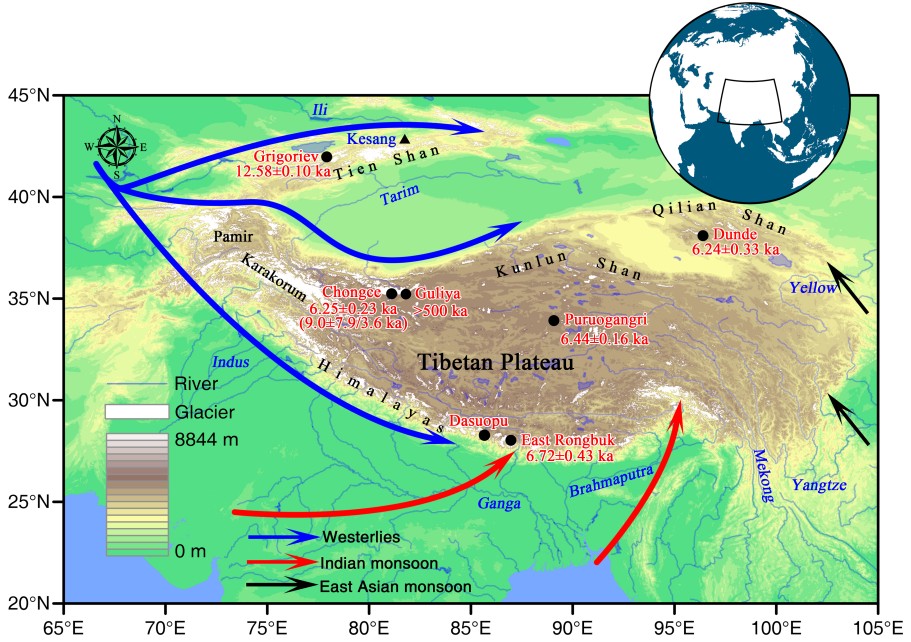

Figure 1: Map showing the locations of ice core drilling sites. The numbers for each

site except Guliya are the measured oldest [14]C ages, while the number inside the

bracket below the Chongce site is the estimated ice age at the ice-bedrock contact.

The schematic positions of the westerlies and the monsoon circulations are from ref.

23. Data of glaciers are from the Global Land Ice Measurements from Space (GLIMS,

available at http://www.glims.org). The topographic data were extracted using

ETOPO1 elevations global data, available from National Oceanic and Atmospheric

Administration at http://www.ngdc.noaa.gov/mgg/global/global.html.



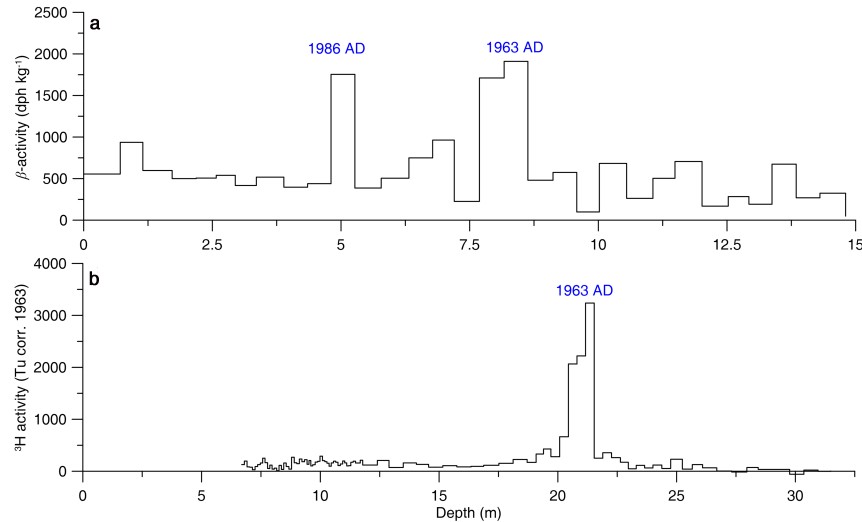

Figure 2: The *β*-activity profile of the Chongce 58.8 m Core 3 (a) and the tritium

profile of the Chongce 216.6 m Core 4 (b). TU (tritium units) is one tritium

atom/1018 hydrogen atoms.





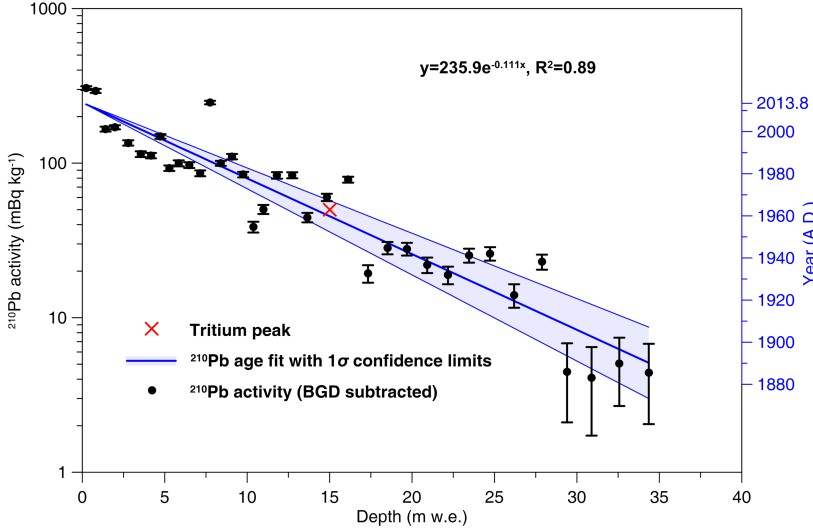

Figure 3: $^{210}$Pb activity profile of the Chongce 216.6 m Core 4 and the derived

age-depth relationship. The $^{3}$H fallout horizon indicating the year 1963 A.D. is

located within the uncertainty of the $^{210}$Pb results. Please note that the 1σ confidence

band is related to the right hand y-axis only.





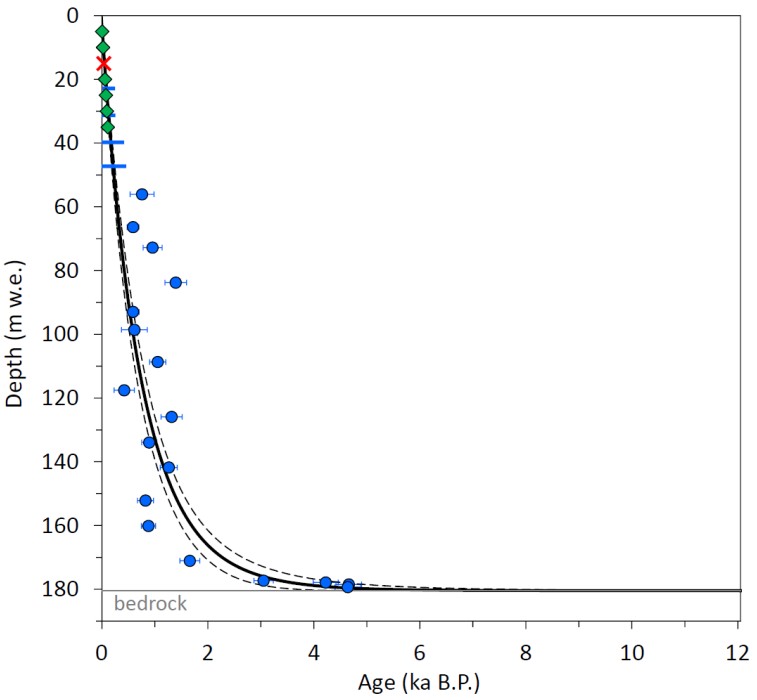

Figure 4: The depth-age relationship of the Chongce 216.6 m Core 4. The dashed

lines represent the 1σ confidence interval of the 2p model fit (solid line). The red

cross stands for the tritium horizon, green diamonds for the [210]Pb ages calculated at

intervals of 5 m w.e. (Fig. 3), and the blue dots for the cal. [14]C ages with 1σ error bar.





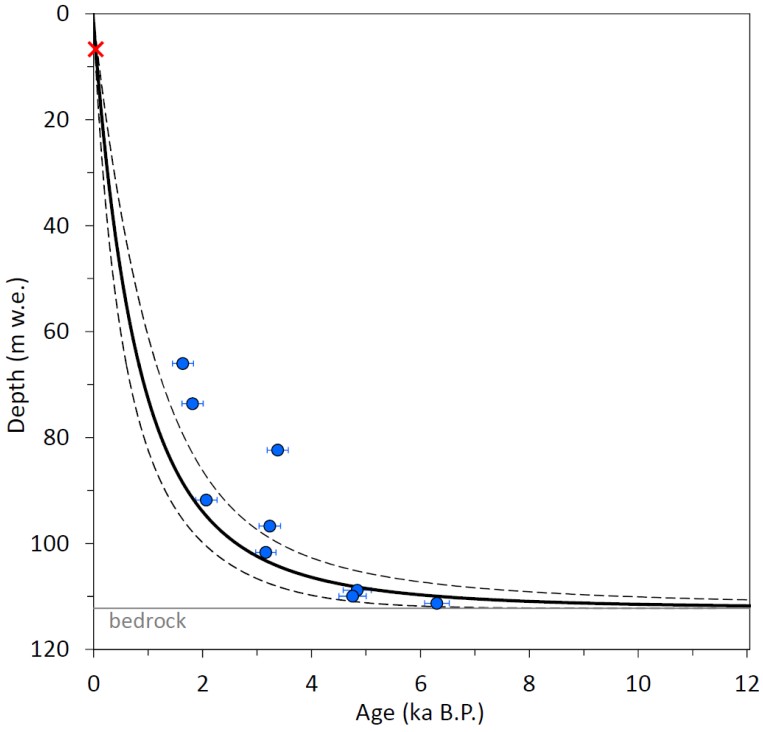


Figure 5: The poorly constrained depth-age relationship of the Chongce 135.8 m Core

2. The dashed lines represent the 1σ confidence interval of the 2p model fit (solid

line). The red cross stands for the *β*-activity horizon (Fig. 2) and the blue dots for the

cal. $^{14}$C ages with 1σ error bars.



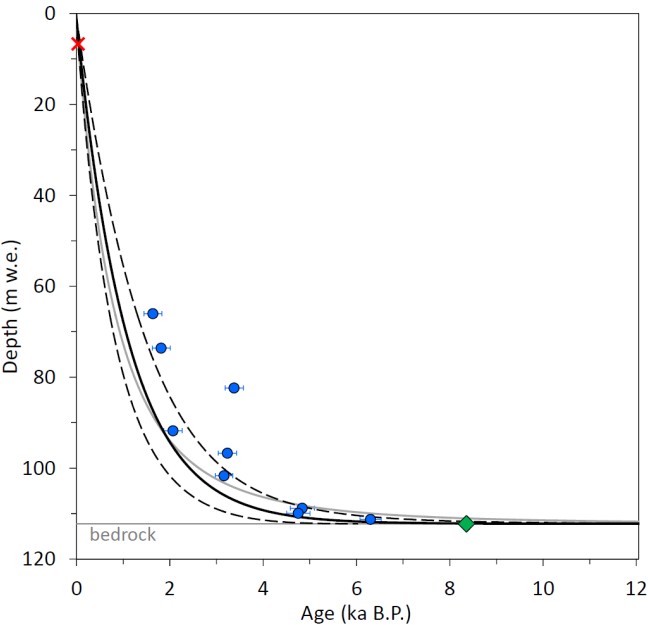


Figure 6: The depth-age relationship of the Chongce 135.8 m Core 2 using additional

age constraint (i.e., age at bedrock estimated from the Chongce 216.6 m Core 4, green

diamond). The dashed lines represent the 1σ confidence interval of the 2p model fit

(solid line). The red cross stands for the $\beta$-activity horizon (Fig. 2) and the blue dots

for the cal. [14]C ages with 1σ error bars. The grey line indicates the depth-age

relationship derived without additional bottom age constraint. Please note that the [14]C

data are all above the fitted line except the deepest point (green diamond) due to the

strong thinning close to ice-bedrock contact.