# Peer review of "Age ranges of the Tibetan ice cores with emphasis on the Chongce ice cores,"

_The Cryosphere, 2018_

## Referee Comment (RC1) · Anonymous Referee #1 · 24 Apr 2018

Dear Editor,

this manuscript brings new valuable information about the Chongse glacier ice core extracted in Tibet that can be published in TC by making major corrections with a neutral review and comparison with other tibetan sites. However, the author needs to clarify his objectives. If through the information of Chongse, the author seeks to discredit the information extracted from other sites like that from Guliya ice core without bringing new, precise and concrete information on this same site, because Guliya doesn't fit with his conclusion, the manuscript has to be rejected.

General comments

In this manuscript, Hou et al. presents preliminary results of dating aa new ice cores extracted from the Chongse glacier in Tibet. Based on three radiogenic dating methods (3H, 210Pb, and 14C), the cores extracted from two different sites on the glacier

would preserve samples dating as far back as the Holocene. This conclusion is mainly based on the deepest samples dating using a robust 14C dating method extracted from WIOC. The extrapolated dating on bedrock age is based on a simple 2-p flow model. The second discussion of this manuscript concerns the comparison of the various ice cores extracted in this Himalayan region and in particular the maximum age preserved in these archives. The author argues that since the majority of these archives, including Chongse, do not preserve more than Holocene ice, the particular site of Guliya displaying 700ky must be questioned.

General comments on both main manuscript sections:

1) Chongse ice cores dating: Chongse ice cores were dated using three different radiogenic methods, 3H, 210Pb and 14C covering different age scales with their own limitations. All methods were applied and results used in an adequate way for their interpretation. 14C method based on WIOC extraction allows to date quasi continuously over the oldest part of the core, reaching an age of 7.0ka ten centimeters above bedrock for deepest core (C4). At this stage of the dating, the use of a very simplistic model 2-p does not bring complementary information in view of the uncertainties related to the parametrization like constant accumulation over time. Using the maximum age of C4 to constrain the C2 model is not appropriate. What seems to be ignored in the manuscript concerns the difference in 14C age obtained between C4 and C2 at intermediate depths, between 2 and 4 ka on C2 (60-110m we) whereas they have rather 0.5-2ka on C4 (80-175m we). These data would rather show that there is no coherence between the two cores and that their records are discontinuous. Considering the data that appears to be available (surface and bedrock topography, surface flow velocity, temperature profile), the application of a 2D or 3D flow model should help to constrain the conclusions on the dating of ice layers.

2) Tibetan ice cores comparison: This paragraph summarizes the maximum ages obtained from the different sites of this region. In order to get a better idea of these points, it is necessary to provide a better description of each of the studied sites (topography,

bedrock temperature, flow) and to describe which methods were applied for the core dating. Special attention seems to be given to the Guliya site, which has an age of up to 700ka and is exceptional for this region. It is not acceptable in this discussion that the author calls into question Guliya's record without providing any additional new information on Guliya glacier, only based on the conclusion of a nearby and very different Chongse glacier.

Specific comments

Title: Not adapted to the content, too general. Suggestion "Chongse ice core, a contribution to Tibetan ice core age review"

Abstract: To review according to the objectives of the manuscript (see introductive comment to editor).

2) Chronology of previous ice cores": Dunde ice core: explain how original chronology was established and what led to the new dating.

Guliya ice core: if this record seems to disturb the author, it will be necessary to provide explanations and more complete arguments. Yao (1997) indicates ages at different depths (AC 1700 at 120m, LGM at 170m, 110ka at 268m) and explain how they were established. It should be noted at this point that the age of Guliya at 170m is of the same order of magnitude as that of Chongse for the same depth, but that Guliya has 100m of additional older ice. The deep dating of Guliya is established by a simple model passing by dated points (36Cl instead of 14C), the technique is similar to that proposed in this manuscript for Chongse, with just as many uncertainties. Even if the last point of 36Cl is very uncertain because of a low chloride concentration, the fact remains that the complete record could only reach 300ka. But in any case, an indisputable point is that the record of Guliya is more than 110ka and calls into question the inappropriate discussions on this point in this manuscript with inexistent ice older than Holocene in Tibetan region.

Dasuopu ice core: explain how this core was dated? Give references like Yao (2002), on CH4 comparison, $\delta$18O depletion missing. . .

Gregoriev ice core: according to Takeuchi (2014), there is in important difference between deepest ice core age (8.155 ka at 81.54m) and soil (12.656 ka). Take care on that potential important difference also on Chongse in your discussion.

Other cores are available, like from Geladaindong (0.5ka at 110m, total core 147m). Check all available data if you want to compare all regional ice core archives.

3) Chongce ice core What are the other characteristics of the Chongse site and ice cores? Density and temperature profiles useful for dating and modelling? From temperature profile, estimate the bedrock potential melting.

4) Measurements Provide a depth distribution table for the 14C samples. How deep were the last samples, how far from bedrock?

For 3H and 210Pb methods, provide uncertainties and references.

5) results About $\beta$-activity profile, how are you sure about the date of each peak without additional information? Did you use annual layer counting to theses depth? What do you know about surface age, did any annual layers disappeared?

For 3H profile in figure 2b, it seems for me that the plotted values are corrected from decay to reach such high numbers? 3237TU for the 1963 peak value is very high in comparison with other ice core data, mostly around few hundreds TU today value for the 1963 peak, see Gelaidaindong core (Kang, 2015). Give reference on the identification of maximum peak.

14C data interpretation: can you argue why in C4, 14C dates are only around 1-2ka for the intermediate depth (80-170m) and between 2-4ka in C2 at 60-110m? For me C4 is discontinuous, may be because of the particular bedrock topography (presence of bedrock canyon), with recent ice on the 0-110m over older dead ice.

The use of a simplistic 2-p model is not appropriate for this kind of site where you know that the accumulation is not constant through the Holocene, but especially to be used on the bottom of the glacier where in principle its exponential approach is unrealistic. You have enough 14C data to trigger glacier bottom age using a simple exponential regression down to bedrock depth for C4.

Lines 234 and 245, you indicate tritium when it is from $\beta$-activity data.

Dating of C2: you are absolutely trying to match a curve from an unsuitable model with data from real measurements, and in addition you add values from other sites to try to force your conclusions. This reasoning and this method are not acceptable. Given the weakness of the model, it is reasonable to remove it from this manuscript.

6) Discussion This chapter is to be reviewed in its entirety. The author must clearly state the objectives of this manuscript. If it comes to presenting new results from the Chongse Core, in this case with major corrections these results could be published. If it's a matter of depressing previous works like those on Guliya glacier, as this manuscript does not bring any innovative data to this site, not the least substantiated discussion, there is no reason to publish it in TC.

On that stage of the review, various technical corrections can wait for a revised version of the manuscript.

---

## Short Comment (SC1) · 24 Apr 2018

Comments: Hou, et al. Age of Tibetan ice cores by L.G. Thompson

The authors raise an important point about the age of the Guliya record as it is unusual in light of previous Tibetan Plateau (TP) records from the Dunde ice cap, Dasuopu Glacier (col) and Puruogangri ice cap. For their Chongce ice cores, the authors used a novel method of radiocarbon dating water-insoluble organic carbon (WIOC) particles at the microgram level from carbonaceous aerosols embedded in the glacier ice from 44 meters to 216.6 meters depth.

They import these dates into a simple two dimensional flow model to develop a time-depth profile. Questions should be raised concerning the use of such a model, given the complex flow regimes in these western Kunlun glaciers. More importantly the authors do not show any climate record derived from stable isotopes from these ice cores

so that the reader can evaluate the quality and continuity of the record and how it compares to the other TP climate records (especially the 1992 Guliya record). Guliya is unusual in that it shows large and abrupt variations in $\delta18O$ below ∼150 m, which other TP records do not contain. Does the longest record from Chongce show similar variations?

Another concern about the Chongce record is geophysical in nature. In 1991 Chinese scientists published a Quaternary Glacial Distribution Map of the TP. According to this map, the terminal moraines around the Guliya ice cap are very close to their maximum position during the last two glaciations. However, this is not the case for the Chongce ice cap located just ∼30 km to the west. Chongce shows the greatest variations in ice extent of any of the ice caps in this region. In addition, the Chongce glacier, which flows from the Chongce ice cap, surged between 1992 and 2014 while the Guliya ice cap remained static (Yasuda and Furuya, 2015; Fig. 3). Therefore, it might be inaccurate to assume that the timescale developed for the Chongce cores should reflect that of Guliya. In light of the evidence indicating the instability of the Chongce's ice flow, the longest core drilled in the deepest section of a valley glacier which flows through a bedrock trough (fig. S2) is very unlikely to be an optimal site for retrieving an undisturbed paleoclimate record. In light of the geophysical considerations discussed above it would be premature to conclude that these results invalidate the much longer Guliya timescale.

Note that the location shown for the 1992 Guliya core in Fig. S1 is incorrect. This should be corrected.

We strongly agree with the authors that more effort is necessary to explore multiple dating techniques to confirm the ages of the TP glaciers, including those from Chongce and Guliya. The Western Kunlun region, located at the intersection of the regions dominated by the westerlies and the SW Monsoon, is climatologically complex and the interactions of multiple air masses makes stable isotope interpretation challenging. Multiple aerosol sources also complicate the reconstruction of the paleo-environmental

records preserved in these ice fields.

Yasuda, T. and Furuya, M. (2015) Dynamics of surge-type glaciers in West Kunlun Shan, Northwestern Tibet. J. Geophys. Res.-Earth Surface, https://doi.org/10.1002/2015JF003511

Please also note the supplement to this comment:
https://www.the-cryosphere-discuss.net/tc-2018-55/tc-2018-55-SC1-supplement.pdf
* * *

---

## Author Comment (AC1) · 30 Apr 2018

Dear Prof. Lonnie Thompson,

Many thanks for your constructive comments. Below a point-to-point response to the comments. The comments are in black, and our response is in blue.
* * *
Comments: Hou, et al. Age of Tibetan ice cores by L.G. Thompson

The authors raise an important point about the age of the Guliya record as it is unusual in light of previous Tibetan Plateau (TP) records from the Dunde ice cap, Dasuopu Glacier (col) and Puruogangri ice cap. For their Chongce ice cores, the authors used a novel method of radiocarbon dating water-insoluble organic carbon (WIOC) particles at the microgram level from carbonaceous aerosols embedded in the glacier ice from 44 meters to 216.6 meters depth.

They import these dates into a simple two dimensional flow model to develop a time-depth profile. Questions should be raised concerning the use of such a model, given the complex flow regimes in these western Kunlun glaciers. More importantly the authors do not show any climate record derived from stable isotopes from these ice cores so that the reader can evaluate the quality and continuity of the record and how it compares to the other TP climate records (especially the 1992 Guliya record). Guliya is unusual in that it shows large and abrupt variations in $\delta^{18}O$ below ~150 m, which other TP records do not contain. Does the longest record from Chongce show similar variations?

Response: We agree that the two dimensional flow model, though widely used for establishing the ice core chronology including the Dunde (Thompson et al., 1989) and the Puruogangri (Thompon et al., 2006) ice cores, is too simple to account for the complex flow regimes close to the glacier bedrock. We will therefore remove the extrapolation from the oldest (and deepest) [14]C data point to the bedrock. Otherwise, we simply used the flow model to fit the dating points for obtaining a continuous age-depth scale. We agree that independent evidence such as the stable isotopes from these ice cores would be helpful to support the dating. We just finished the measurement of stable isotopes of the Chongce 135.8 m Core 2 and the 58.8 m Core 3 and they suggest Holocene origin in agreement with our dating. We are preparing the corresponding manuscript and therefore cannot post these $\delta^{18}O$ profiles here and make them public at this moment. Nevertheless, we will provide a copy of these $\delta^{18}O$ profiles for the review purpose only.

Another concern about the Chongce record is geophysical in nature. In 1991 Chinese scientists published a Quaternary Glacial Distribution Map of the TP. According to this map, the terminal moraines around the Guliya ice cap are very close to their maximum position during the last two glaciations. However, this is not the case for the Chongce ice cap located just ~30 km to the west. Chongce shows the greatest variations in ice extent of any of the ice caps in this region. In addition, the Chongce glacier, which flows from the Chongce ice cap, surged between 1992 and 2014 while the Guliya ice cap remained static (Yasuda and Furuya, 2015; Fig. 3). Therefore, it might be inaccurate to assume that the timescale developed for the Chongce cores should reflect that of Guliya. In light of the evidence indicating the instability

of the Chongce's ice flow, the longest core drilled in the deepest section of a valley glacier which flows through a bedrock trough (fig. S2) is very unlikely to be an optimal site for retrieving an undisturbed paleoclimate record. In light of the geophysical considerations discussed above it would be premature to conclude that these results invalidate the much longer Guliya timescale.

Response: Thank you for bringing to our attention of Yasuda and Furuya's work on the dynamics of surge-type glaciers in West Kunlun Shan. From Fig. 3 of their paper, it is clear that the surged area is confined within the Chongce glacier (Fig. 1). Using topographical maps, Shuttle Radar Topography Mission (SRTM) and Landsat data, we have examined the area changes of glaciers on the Western Kunlun Mountain (including the Chongce and Guliya ice caps) since the 1970s (Fig. 1. Wang, Y., Hou, S., Huai, B., An, W., Pang, H., Liu, Y.: Glacier anomaly over the Western Kunlun Mountains, northwestern Tibetan Plateau, since the 1970s, *J. Glaciol.*, 3[rd] revision). For the whole area, change of the glacier area reveals insignificant shrinkage by $0.07 \pm 0.1\%$ yr$^{-1}$ from the 1970s to 2016. The Chongce glacier retreated between 1977 and 1990, and advanced from 1990 to 2011 (period of surge), then remained stable until 2016. In contrast, the Chongce ice cap remained static from the 1977 to 2016 (Fig. 1), confirming the stability of the ice cap where our ice cores were recovered. Moreover, we observed similar mass changes of surge-type and non-surge-type glaciers over the Western Kunlun Mountains, suggesting that the flow instabilities seem to have little effect on the glacier-wide mass balance. Similar mass budgets for surging and non-surging glaciers have also been reported in the Pamirs and Karakoram (Gardelle et al., 2013). Based on the estimate of elevation changes over the West Kunlun Mountain by Lin et al. (2017) and Zhou

et al. (2018) (Figs 3 and 4), it is reasonable that Chongce ice cap is in balance between 1973

and 2014. Therefore, the impact of surge over the Chongce glacier is minimal, if any, on the

stratigraphy of the Chongce ice cap, especially in its accumulation zone where our Chongce

ice cores were drilled.

[Figure]

Fig.1. Map showing the Chongce Ice Cap (CIC) and the Chongce Glacier (CG), with the

terminus positions at different time. The star shows the position of the drilling site of the

Chongce Cores 2 and 3, which might be an optimal site for retrieving an undisturbed

paleoclimate record. The inset is from Fig. 3 of Yasuda and Furuya (2015) with the red area

showing the surged area confined within the Chongce glacier. Terminus positions are

determined from Landsat images as shown in Fig. 2.

[Figure]

Landsat MSS (17, Feb, 1977)        Landsat TM ( 15, Nov. 1990)

Landsat TM ( 5, Aug. 2011)        Landsat 8 ( 5,Oct. 2016)

Fig. 2. Data for Chongce glacier and ice cap terminus position assessment. They are
co-registered to the topographical maps and the accuracy of co-registration is about 20 m
(slightly more than half of one pixel of Landsat images)

[Figure]

Fig.3. Glacier height changes from 2000 to the 2010s from Lin et al. (2017)

[Figure]

Fig.4. Glacier elevation change from 1973 to 2010 from Zhou et al. (2018)

Although the 1991 Quaternary Glacial Distribution Map of the TP (Li and Li, 1991) can provide valuable information about the quaternary glacier variation on the TP, its spatial resolution (1:3 000 000) may be insufficient to delineate the variation features of a specific glacier or ice cap. Later, Jiao et al. (2000) studied the evolution of glaciers in the West Kunlun Mountains during the past 32 ka (Fig.5). It is clear that the Chongce glacier advanced much during the LGM, while the terminal moraines in front of the Chongce ice cap are very close to their maximum position during the LGM, similar to those around the Guliya ice cap, confirming the stability of the Chongce ice cap since the LGM.

[Figure]

Fig.5. Map showing the glacier distribution and the lower limit of the LGM in the West Kunlun Mountains (Jiao et al., 2000). CIC: Chongce Ice Cap, CG: Chongce Glacier, GIC: Guliya Ice Cap. 1: present glacier, 2: terminal moraine during the LGM, 3: terminal moraine during the Neoglaciation, 4. lakes

Note that the location shown for the 1992 Guliya core in Fig. S1 is incorrect. This should be corrected.

Response: We used the coordinates (35°17' N, 81°29' E) published in Thompson et al. (1997) for the location of the 1992 Guliya core. We would be happy to make a correction if more accurate coordinates are provided.

We strongly agree with the authors that more effort is necessary to explore multiple dating techniques to confirm the ages of the TP glaciers, including those from Chongce and Guliya. The Western Kunlun region, located at the intersection of the regions dominated by the westerlies and the SW Monsoon, is climatologically complex and the interactions of multiple air masses makes stable isotope interpretation challenging. Multiple aerosol sources also complicate the reconstruction of the paleo-environmental records preserved in these ice fields.

Response: We fully agree with this comment. Thanks to the new analytical techniques, we should have more opportunities to decipher the complexity of the Tibetan ice core records.

References

Gardelle, J., Berthier, E., Arnaud, Y., and Kääb A.: Region-wide glacier mass balances over the Pamir–Karakoram–Himalaya during 1999–2011, The Cryosphere, 7, 1263–1286, doi: 10.5194/tc-7-1263-2013, 2013.

Jiao, K., Yao, T., and Li, S.: Evolution of glaciers and environment in the West Kunlun Mountains during the past 32 ka, J. Glacio. Geocryo., 22, 250-256, 2000 (in Chinese with English abstract).

Li, B., and Li, J.: Quaternary glacial distribution map of Qinghai-Xizang (Tibet) Plateau. Science Press, Beijing, 1991.

Lin, H., Li, G., Cuo, L., Hooper, A., and Ye, Q.: A decreasing glacier mass balance gradient from the edge of the Upper Tarim Basin to the Karakoram during 2000–2014. Sci. Rep., 7, 612, doi:10.1038/s41598-017-07133-8, 2017.

Thompson, L. G., Mosley-Thompson, E., Davis, M., Bolzan, J., Dai, J., Klein, L., Yao, T., Wu, X., Xie, Z., and Gundestrup, N.: Holocene-late pleistocene climatic ice core records from Qinghai-Tibetan Plateau, Science, 246, 474-477, doi:10.1126/science.246.4929.474, 1989.

Thompson, L. G., Yao, T., Davis, M. E., Henderson, K. A., Mosley-Thompson, E., Lin, P.-N., Beer, J., Synal, H.-A., Cole-Dai, J., and Bolzan, J.F.: Tropical climate instability: the last glacial cycle from a Qinghai-Tibetan ice core, Science, 276, 1821-1825, doi: 10.1126/science.276.5320.1821, 1997.

Thompson, L. G., Yao, T., Davis, M., Mosley-Thompson, E., Mashiotta, T., Lin, P., Mikhalenko, V., and Zagorodnov, V.: Holocene climate variability archived in the Puruogangri ice cap on the central Tibetan Plateau, Ann. Glaciol., 43, 61-69, doi:10.3189/172756406781812357, 2006.

Yasuda, T. and Furuya, M.: Dynamics of surge-type glaciers in West Kunlun Shan, Northwestern Tibet, J. Geophys. Res. Earth Surf., 120, 2393–2405, doi: 10.1002/2015JF003511, 2015.

Zhou, Y., Li, Z., Li, J., Zhao, R., and Ding, X.: Glacier mass balance in the Qinghai–Tibet Plateau and its surroundings from the mid-1970s to 2000 based on Hexagon KH-9 and SRTM DEMs, Remote Sens. Environ., 210, 96-112, doi: 10.1016/j.rse.2018.03.020, 2018.

---

## Author Comment (AC2) · 16 May 2018

Dear Referee,

Many thanks for your constructive comments. Below a point-to-point response to the comments. The comments are in black, and our response is in blue.
* * *
This manuscript brings new valuable information about the Chongse glacier ice core extracted in Tibet that can be published in TC by making major corrections with a neutral review and comparison with other tibetan sites. However, the author needs to clarify his objectives. If through the information of Chongse, the author seeks to discredit the information extracted from other sites like that from Guliya ice core without bringing new, precise and concrete information on this same site, because Guliya doesn't fit with his conclusion, the manuscript has to be rejected.

Response: We thank the reviewer for the positive viewpoint of our results. We have to say that, from the beginning, it is beyond our intention to discredit the chronology of the Guliya ice core. In fact, we had initially been inspired by the exceptional length of the Guliya core to drill the nearby Chongce ice cores. This is also the reason for us to collect a total of 22 samples from the longest Chongce core (216.6 m Core 4) for $^{14}$C measurement, by sampling from as shallow as 30.49 m depth to the bottom (0.57 m above the ice-bedrock contact). When we got the cal. $^{14}$C ages of these 22 samples, we were puzzled by the results that the measured oldest cal. $^{14}$C age is only 4.59±0.24 ka B. P.. Moreover, several samples from the upper ice core sections are, in fact, not ideal for $^{14}$C measurements due to their relatively

young ages of around ~1 ka or younger (see discussion in lines 203-214 and Fig. 4 in our TCD manuscript). Afterwards, we collected 9 samples from the Chongce 135.8 m Core 2 by sampling from 79.46 m depth to the bottom (0.77 m above the ice-bedrock contact). Although the measured oldest cal. $^{14}$C age (6.25±0.23 ka B. P.) is slightly older than that of Core 4, it is apparent that the measured oldest cal. $^{14}$C ages for both the Chongce Core 2 and Core 4, as well as other Tibetan ice cores except Guliya (i.e., Dunde, Puruogangri, East Rongbuk, Fig. 1), fall within a similar age range.

To meet the reviewer's comments for "a neutral review and comparison with other tibetan sites", we would like to change the title from "Age of the Tibetan ice cores" to "Age ranges of the Tibetan ice cores with emphasis on the Chongce ice cores, western Kunlun Mountains". We would also like to present, in a neutral way, the results of the Chongce ice cores, the results of other Tibetan ice cores from their original literatures, the remarkable age range difference of the Guliya and other Tibetan ice cores, and finally to conclude that "more effort is necessary to explore multiple dating techniques to confirm the ages of the TP glaciers, including those from Chongce and Guliya" (Thompson, 2018a).

[Figure]

Fig. 1. Map showing the locations of ice core drilling sites. The numbers for each site except Guliya are the measured oldest [14]C ages, while the number inside the bracket below the Chongce site is the estimated ice age at the ice-bedrock contact. The schematic positions of the westerlies and the monsoon circulations are from Yao et al. (2013). Data of glaciers are from the Global Land Ice Measurements from Space (GLIMS, available at http://www.glims.org). The topographic data were extracted using ETOPO1 elevations global data, available from National Oceanic and Atmospheric Administration at http://www.ngdc.noaa.gov/mgg/global/global.html.

General comments

In this manuscript, Hou et al. presents preliminary results of dating aa new ice cores extracted from the Chongse glacier in Tibet. Based on three radiogenic dating methods ([3]H, [210]Pb, and

$^{14}$C), the cores extracted from two different sites on the glacier would preserve samples dating as far back as the Holocene. This conclusion is mainly based on the deepest samples dating using a robust $^{14}$C dating method extracted from WIOC. The extrapolated dating on bedrock age is based on a simple 2-p flow model. The second discussion of this manuscript concerns the comparison of the various ice cores extracted in this Himalayan region and in particular the maximum age preserved in these archives. The author argues that since the majority of these archives, including Chongse, do not preserve more than Holocene ice, the particular site of Guliya displaying 700ky must be questioned.

General comments on both main manuscript sections:

1) Chongse ice cores dating: Chongse ice cores were dated using three different radiogenic methods, $^3$H, $^{210}$Pb and $^{14}$C covering different age scales with their own limitations. All methods were applied and results used in an adequate way for their interpretation. $^{14}$C method based on WIOC extraction allows to date quasi continuously over the oldest part of the core, reaching an age of 7.0 ka ten centimeters above bedrock for deepest core (C4). At this stage of the dating, the use of a very simplistic model 2-p does not bring complementary information in view of the uncertainties related to the parametrization like constant accumulation over time. Using the maximum age of C4 to constrain the C2 model is not appropriate. What seems to be ignored in the manuscript concerns the difference in $^{14}$C age obtained between C4 and C2 at intermediate depths, between 2 and 4 ka on C2 (60-110m we) whereas they have rather 0.5-2ka on C4 (80-175m we). These data would rather show that there is no coherence between the two cores and that their records are discontinuous.

Considering the data that appears to be available (surface and bedrock topography, surface flow velocity, temperature profile), the application of a 2D or 3D flow model should help to constrain the conclusions on the dating of ice layers.

Response: We agree that the two dimensional flow model, though widely used for establishing the ice core chronology including the Dunde (Thompson et al., 1989) and the Puruogangri (Thompson et al., 2006) ice cores, is limited and cannot account for the complex flow regimes close to the glacier bedrock. We have to point out that it is always a challenge to model the glacier bottom regime very close to bedrock, and the application of a more complex 2D or 3D glacier flow model faces a similar difficulty for dating the glacier bottom ice due to its very complex behavior without constraints (e.g. Luthi and Funk, 2000).

In our approach, the flow model is simply used to obtain a continuous age-depth scale by fitting to the dating points. Regarding the available age constraints and the uncertainties of the individual dating points we refrained from applying a more sophisticated fitting procedure that would allow, for instance, changes in accumulation over time in order not to over fit (or over interpret) the available data points. Based on the chosen approach, the age constrained 2p-model results in a reasonable fit. The derived average annual accumulation rate is similar to the values obtained from the $^3H/\beta$-activity peak and the $^{210}Pb$ dating methods. This suggests that the underlying assumptions of this approach seem reasonable. In any case, more independent constraints are needed to improve the present results.

Regarding the extrapolation to the very bedrock and the constrain of the age model for Core 4 using information from Core 2, we fully agree, that this approach is far from satisfactory, as we already discussed in the manuscript. We decided to use this additional constraint in order

to get the most reasonable age estimate combining all available dating information we have got from both Chongce cores. The limitation of this approach was appropriately reflected in the large uncertainties of the age estimates and discussed accordingly. However, we do understand the reviewer's concerns. Since we have cal. $^{14}$C ages very close to the bottom of both cores, which are in agreement with each other and in-line with the general discussion and conclusions of the paper, we thus decided to remove this approach from the revised version of the manuscript as no significant loss in information will result. Nevertheless, to make full use of the information available, this part will instead be moved to the supplement. In any case, independent evidence such as stable isotopes might be very helpful, which is under way for the Chongce ice cores.

The incoherence between Core 2 and Core 4 may be caused by their surface topography, resulting in different accumulation rate at their drilling sites. Based on the $\beta$-activity reference of 1963 AD, the mean annual accumulation rate of Core 3 (several meters away from Core 2 drilling site) is calculated to be 140 mm w.e. (water equivalent) /year for the period of 1963–2012 AD, while the mean annual accumulation rate of Core 4 is 297 mm w.e./year for the period of 1963-2013 AD. It is possible that the Core 4 drilling site may receive extra snow supplies, such as snow drifting, whereas part of the snow deposition at the Core 3 drilling site may be blown away due to wind scouring (Fisher et al., 1983). The impact of wind scouring on the ice core drilled from the Dasuopu summit was also suggested (Thompson et al., 2018b). Nevertheless, a full understanding of this difference will require a long-term *in situ* observation that is unavailable at this moment. Further, a possible explanation for the determined ages at intermediate depths of Core 2 was discussed in the

manuscript (lines 203-214). Generally for the [14]C dating by WIOC, it is not recommended to discuss individual dating points. With the accumulation of Core 4 being approximately twice as high compared to Core 2, the ages of around ~1.7±0.2 ka for Core 4 seem to be not perfect but still in agreement with the ages determined for Core 2 when considering the uncertainties. The single data point of ~3.3±0.2 ka in Core 4 is indeed outside the expected range but as stated above, it should be avoided to overrate the relevance of a single data point considering all uncertainties and potential bias from sample composition and preparation in this highly sensitive analytical approach.

2) Tibetan ice cores comparison: This paragraph summarizes the maximum ages obtained from the different sites of this region. In order to get a better idea of these points, it is necessary to provide a better description of each of the studied sites (topography, bedrock temperature, flow) and to describe which methods were applied for the core dating. Special attention seems to be given to the Guliya site, which has an age of up to 700ka and is exceptional for this region. It is not acceptable in this discussion that the author calls into question Guliya's record without providing any additional new information on Guliya glacier, only based on the conclusion of a nearby and very different Chongse glacier.

Response: We will include the following basic information for each of the ice cores in the manuscript, and encourage the readers to find details in their respective references. For a neutral review of the result, we will delete the discussion for calling into question Guliya's chronology, but just reminding the remarkable temporal range difference of the Guliya and other Tibetan ice cores.

We notice that, in 2015, a new Guliya ice core to bedrock (309.73 m in length) was drilled close to the location of the 1992 Guliya core drill site, as well as three cores to bedrock (50.72 m, 51.38 m, 50.86 m in length) from the Guliya Summit. Thompson et al. (2018b) suggested that future analyses will include $^{14}$C on organic material trapped in the ice, and $^{36}$Cl, beryllium-10 ($^{10}$Be), $\delta^{18}$O of air in bubbles trapped in the ice, and argon isotopic ratios ($^{40}$Ar/$^{38}$Ar) on deep sections of the 309.73 m core to determine more precisely the age of the ice cap. We look forward to their new results.

Information for the ice cores that will be included in the revised manuscript:

The Guliya ice core: In 1992, a 308.6 m ice core to bedrock was recovered at an elevation of 6200 m a.s.l. from the Guliya ice cap (35°17′ N, 81°29′ E) on the northwestern TP (Fig. 1). The Guliya ice cap is surrounded by vertical ice walls 30 to 40 m high and has internal temperatures of -15.6 °C at 10 m, -5.9 °C at 200 m, and -2.1 °C at its base. Top 266 m of the Guliya core was dated to a period spanning 110 ka B.P., and the ice below 290 m was suggested to be >500 ka B.P., or to ~760 ka B.P. at the ice-bedrock contact based on $^{36}$Cl-dead ice at the bottom section (Thompson et al., 1997).

The Dunde ice cores: In 1987, three ice cores to bedrock (139.8 m, 136.6 m and 138.4 m in length) were recovered at an altitude of 5325 m a.s.l from the Dunde ice cap (38°06′ N, 96°24′ E) in the Qilian Shan on the northern TP (Fig. 1). Surface and basal borehole temperatures were -7.3 °C and -4.7 °C, respectively. The 2 ‰ shift in $\delta^{18}$O, concurrent with a sudden increase in dust concentration 14 m above the bedrock was interpreted as evidence of glacial-stage ice (Thompson et al., 1989). The core was extrapolated to 40 ka B.P. at the

depth of 5 m above the bedrock by applying a two dimensional flow model, and was suggested to be potentially more than 100 ka B.P. at the ice-bedrock contact (Thompson et al., 1989). Later, Thompson et al. (2005) provided a single [14]C date of 6.24±0.33 ka B.P. for a sample collected close to the ice–bedrock contact, and suggested the possibility that this core may be of Holocene origin.

The Puruogangri ice cores: In 2000, three ice cores (118.4 m, 214.7 m and 152 m in length) were recovered at an altitude of 6070 m a.s.l. from the Puruogangri ice cap (33° 55′ N, 89° 05′ E) on the central TP (Fig. 1). Borehole temperatures were –9.7 °C at 10 m and –5 °C at the ice/bedrock contact. The measured oldest [14]C date is 6.44±0.16 ka B.P. at 210.5 m depth of the 214.7 m ice core. The dating was extrapolated another 0.5 m further down to 7 ka B.P. (Thompson et al., 2006). The Puruogangri ice cores may be of Holocene origin (Thompson et al., 2005).

The Dasuopu ice cores: In 1997, three ice cores were drilled from the Dasuopu glacier (28° 23′ N, 85° 43′ E) in the Himalayas. The first core (159.9 m in length) was drilled at an altitude of 7000 m a.s.l., and two more cores (149.2 and 167.7 m in length, respectively) were drilled to bedrock 100 m apart on the col at an altitude of 7200 m a.s.l. (Thompson et al., 2000). Borehole temperatures were -16 °C at 10 m and –13 °C at the ice/bedrock contact (Yao et al., 2002). The $\delta^{18}O$ record of the Dasuopu ice core lacks the 5 to 6 ‰ depletion that characterises glacial stage ice from the tropics to the polar regions (Yao et al., 2002). Furthermore, Dasuopu's basal ice does not contain as low as 0.4 ppmv (parts per million by volume) methane levels that characterise glacial ice in polar ice cores (Raynaud et al., 2000).

Thus, it was suggested that the Dasuopu ice field accumulated entirely during the Holocene (Thompson et al., 2005).

The Grigoriev ice core: In 2007, an ice core to bedrock (86.87 m long) was recovered at an altitude of 4563 m a.s.l at the top of the Grigoriev ice cap (41° 59′ N, 77° 55′ E) in the west Tien Shan (Fig. 1). Borehole temperatures were was -2.7 °C at 10 m and –3.9 °C at the ice/bedrock contact. Takeuchi et al. (2014) suggested that the bottom age of the Grigoriev ice core coincides with the Younger Dryas cold period (YD, 11.7 - 12.9 ka B.P.). However, the oldest $^{14}$C age (12.58±0.10 ka) is obtained from a soil sample collected underneath the glacier, which should be considered as an upper constraint for the age of ice at the ice-bedrock contact. A close examination of the Grigoriev ice core $\delta^{18}$O profile (Fig. 4 in Takeuchi et al., 2014) reveals that the lowest $\delta^{18}$O values occurred ~8 ka B.P., between two cal. $^{14}$C dates of 8.00±0.04 ka B.P. and 8.07±0.06 ka B.P., but the bottom section, corresponding to the age between 8.07±0.06 ka B.P. and 12.58±0.10 ka B.P., shows no further decrease of $\delta^{18}$O below the low values around 8 ka B.P.. Given the high sensitivity of $\delta^{18}$O to temperature for the Grigoriev ice core (Takeuchi et al., 2014), significantly more negative $\delta^{18}$O values towards the bottom section of the Grigoriev core should have been observed if the bottom age indeed reaches YD. Therefore, the Grigoriev bottom ice is very likely younger than YD, and could be deposited during the early Holocene.

Specific comments

Title: Not adapted to the content, too general. Suggestion "Chongse ice core, a contribution to Tibetan ice core age review"

Response: We consider to change the title as "Age ranges of the Tibetan ice cores with emphasis on the Chongce ice cores, western Kunlun Mountains".

Abstract: To review according to the objectives of the manuscript (see introductive comment to editor).

Response: We will delete the last sentence in the abstract "The significant discrepancy between the Guliya and all the other Tibetan ice core chronology calls for a revisit of the Guliya ice core". Instead, we include the following sentence "The remarkable discrepancy between the Guliya and all the other Tibetan ice core chronology implies that more effort is necessary to explore multiple dating techniques to confirm the ages of the TP glaciers, including those from Chongce and Guliya."

2) Chronology of previous ice cores": Dunde ice core: explain how original chronology was established and what led to the new dating.

Response: We will include the following information in the manuscript.

The Dunde ice cores: In 1987, three ice cores to bedrock (139.8 m, 136.6 m and 138.4 m in length) were recovered at an altitude of 5325 m a.s.l from the Dunde ice cap (38° 06′ N, 96° 24′ E) in the Qilian Shan on the northern TP (Fig. 1). Surface and basal borehole temperatures are -7.3 °C and -4.7 °C, respectively. The 2 ‰ shift in $\delta^{18}O$, concurrent with a sudden increase in dust concentration 14 m above the bedrock was interpreted as evidence of glacial-stage ice (Thompson et al., 1989). The core was extrapolated to 40 ka B.P. at the

depth of 5 m above the bedrock by applying a two dimensional flow model, and was suggested to be potentially more than 100 ka B.P. at the ice-bedrock contact (Thompson et al., 1989). Later, Thompson et al. (2005) provided a single $^{14}$C date of 6.24±0.33 ka B.P. for a sample collected close to the ice–bedrock contact, and suggested the possibility that this core may be of Holocene origin.

Guliya ice core: if this record seems to disturb the author, it will be necessary to provide explanations and more complete arguments. Yao (1997) indicates ages at different depths (AC 1700 at 120m, LGM at 170m, 110ka at 268m) and explain how they were established. It should be noted at this point that the age of Guliya at 170m is of the same order of magnitude as that of Chongse for the same depth, but that Guliya has 100m of additional older ice. The deep dating of Guliya is established by a simple model passing by dated points ($^{36}$Cl instead of $^{14}$C), the technique is similar to that proposed in this manuscript for Chongse, with just as many uncertainties. Even if the last point of $^{36}$Cl is very uncertain because of a low chloride concentration, the fact remains that the complete record could only reach 300ka. But in any case, an indisputable point is that the record of Guliya is more than 110ka and calls into question the inappropriate discussions on this point in this manuscript with inexistent ice older than Holocene in Tibetan region.

Response: Many thanks for providing a different perspective. We will simply remind the remarkable discrepancy between the Guliya and all the other Tibetan ice core chronology, and leave the question open.

Dasuopu ice core: explain how this core was dated? Give references like Yao (2002),

on CH$_4$ comparison, $\delta$ $^{18}$O depletion missing**...**

Response: We will include the following information in the manuscript.

The Dasuopu ice cores: In 1997, three ice cores were drilled from the Dasuopu glacier (28°

23′ N, 85° 43′ E) in the Himalayas. The first core (159.9 m in length) was drilled at an

altitude of 7000 m a.s.l., and two more cores (149.2 and 167.7 m in length, respectively) were

drilled to bedrock 100 m apart on the col at an altitude of 7200 m a.s.l. (Thompson et al.,

2000). Borehole temperatures were -16 °C at 10 m and –13 °C at the ice/bedrock contact

(Yao et al., 2002). The $\delta^{18}$O record of the Dasuopu ice core lacks the 5 to 6 ‰ depletion that

characterises glacial stage ice from the tropics to the polar regions (Yao et al., 2002).

Furthermore, Dasuopu's basal ice does not contain as low as 0.4 ppmv (parts per million by

volume) methane levels that characterise glacial ice in polar ice cores (Raynaud et al., 2000).

Thus, it was suggested that the Dasuopu ice field accumulated entirely during the Holocene

(Thompson et al., 2005).

Gregoriev ice core: according to Takeuchi (2014), there is in important difference between

deepest ice core age (8.155 ka at 81.54m) and soil (12.656 ka). Take care on that potential

important difference also on Chongse in your discussion.

Response: We will include the following information in the manuscript.

The Grigoriev ice core: In 2007, an ice core to bedrock (86.87 m long) was recovered at an altitude of 4563 m a.s.l at the top of the Grigoriev ice cap (41° 59′ N, 77° 55′ E) in the west Tien Shan (Fig. 1). Borehole temperatures were was -2.7 °C at 10 m and –3.9 °C at the ice/bedrock contact. Takeuchi et al. (2014) suggested that the bottom age of the Grigoriev ice core coincides with the Younger Dryas cold period (YD, 11.7 - 12.9 ka B.P.). However, the oldest [14]C age (12.58±0.10 ka) is obtained from a soil sample collected underneath the glacier, which should be considered as an upper constraint for the age of ice at the ice-bedrock contact. A close examination of the Grigoriev ice core $\delta^{18}$O profile (Fig. 4 in Takeuchi et al., 2014) reveals that the lowest $\delta^{18}$O values occurred ~8 ka B.P., between two cal. [14]C dates of 8.00±0.04 ka B.P. and 8.07±0.06 ka B.P., but the bottom section, corresponding to the age between 8.07±0.06 ka B.P. and 12.58±0.10 ka B.P., shows no further decrease of $\delta^{18}$O below the low values around 8 ka B.P.. Given the high sensitivity of $\delta^{18}$O to temperature for the Grigoriev ice core (Takeuchi et al., 2014), significantly more negative $\delta^{18}$O values towards the bottom section of the Grigoriev core should have been observed if the bottom age indeed reaches YD. Therefore, the Grigoriev bottom ice is very likely younger than YD, and could be deposited during the early Holocene.

Other cores are available, like from Geladaindong (0.5ka at 110m, total core 147m). Check all available data if you want to compare all regional ice core archives.

Response: There are quite a few ice cores ever drilled from the TP other than what discussed in our manuscript. But most of them didn't reach bedrock. Even for those reaching bedrock, there is no information available about age of their bottom sections (e.g., Geladaindong and

Nyainqêntanglha (Kang et al., 2015)). Therefore, we consider to change the title as "Age ranges of the Tibetan ice cores with emphasis on the Chongce ice cores, western Kunlun Mountains".

3) Chongce ice core What are the other characteristics of the Chongse site and ice cores? Density and temperature profiles useful for dating and modelling? From temperature profile, estimate the bedrock potential melting.

Response: We will include the density profiles of the Chongce ice cores (Fig. 2) and borehole temperature profiles (Fig. 3) in the manuscript.

Borehole temperatures are -12.8 °C, -12.6 °C and -12.6 °C at 10 m for Core 1, Core 2 and Core 3, -8.8 °C and -8.8 °C at 130 m for Core 1 and Core 2, respectively, suggesting that the Chongce ice cap is frozen to its bedrock.

[Figure]

Fig. 2. Density profiles of the Chongce Core 2, Core 3 and Core 4 (see Fig. S2 in the TCD manuscript for location of these ice cores).

[Figure]

Fig. 3. Borehole temperature profiles of the Chongce Core 1, Core 2 and Core 3 (see Fig. S2 in the TCD manuscript for location of these ice cores).

4) Measurements Provide a depth distribution table for the [14]C samples. How deep were the last samples, how far from bedrock?

Response: We will include the following tables for the [14]C measurements in the Supplement. The lowest [14]C samples are 0.77 m (Core 2) and 0.57 m (Core 4) above the bedrock. This information will be included in the revision.

Table 1. The $^{14}$C dating of 216.6 m Chongce ice core. Absolute uncertainties are given as 1σ range.

| Sample # | Depth (m) | Mass (g) | WIOC (μg) | F$^{14}$C | $^{14}$C age (ka B.P.) | Cal. age (ka B.P.) |
|---|---|---|---|---|---|---|
| CC-1 | 30.49-32 | 1178.5 | 49.8±3.8 | 1.00±0.01 | 0.002±0.010 | 0.025-0.26 |
| CC-2 | 40.11-40.97 | 445.7 | 13.6±0.7 | 1.04±0.03 | -0.297±0.236 | 0.013-0.269 |
| CC-3 | 50.06-50.82 | 426.2 | 10.2±0.6 | 0.98±0.04 | 0.148±0.318 | < 0.430 |
| CC-4 | 58.88-59.72 | 464.1 | 22.0±1.2 | 0.97±0.02 | 0.248±0.154 | < 0.470 |
| CC-5 | 69.37-70.09 | 407.5 | 13.2±0.7 | 0.91±0.03 | 0.717±0.241 | 0.513-0.921 |
| CC-6 | 81.41-82.52 | 312.2 | 18.6±1.6 | 0.94±0.01 | 0.537±0.129 | 0.479-0.667 |
| CC-7 | 88.98-90.20 | 345.6 | 30.0±2.7 | 0.89±0.02 | 0.950±0.198 | 0.687-1.055 |
| CC-8 | 102.13-102.85 | 479.2 | 16.6±0.9 | 0.84±0.02 | 1.418±0.193 | 1.090-1.548 |
| CC-9 | 112.51-113.69 | 394.3 | 17.8±1.4 | 0.94±0.02 | 0.537±0.133 | 0.469-0.669 |
| CC-10 | 119.13-120.53 | 386.1 | 17.9±1.5 | 0.93±0.03 | 0.576±0.261 | 0.310-0.781 |
| CC-11 | 131.55-132.32 | 390.5 | 23.1±1.4 | 0.87±0.02 | 1.073±0.154 | 0.800-1.177 |
| CC-12 | 142.33-142.95 | 316.4 | 15.1±1.0 | 0.95±0.03 | 0.374±0.211 | < 0.629 |
| CC-13 | 152.22-152.99 | 411.8 | 16.7±1.0 | 0.85±0.02 | 1.348±0.191 | 1.013-1.516 |
| CC-14 | 161.93-162.66 | 432.2 | 21.6±1.3 | 0.89±0.02 | 0.892±0.154 | 0.679-0.935 |
| CC-15 | 171.35-172.05 | 430.0 | 20.2±1.2 | 0.85±0.02 | 1.303±0.160 | 1.010-1.355 |
| CC-16 | 183.46-184.16 | 411.8 | 20.6±1.2 | 0.91±0.02 | 0.800±0.172 | 0.567-0.920 |
| CC-17 | 192.56-193.32 | 452.5 | 25.7±1.5 | 0.90±0.02 | 0.879±0.148 | 0.684-0.927 |
| CC-18 | 205.38-205.99 | 343.3 | 18.9±1.1 | 0.81±0.02 | 1.673±0.174 | 1.394-1.805 |
| CC-19 | 212.48-213.36 | 511.9 | 28.5±1.6 | 0.70±0.01 | 2.837±0.152 | 2.787-3.156 |
| CC-20 | 213.36-214.10 | 382.6 | 24.6±1.4 | 0.62±0.01 | 3.778±0.167 | 3.930-4.413 |
| CC-21 | 214.10-215.08 | 412.8 | 23.4±1.4 | 0.60±0.01 | 4.100±0.173 | 4.408-4.854 |
| CC-22 | 215.08-216.03 | 367.1 | 23.6±1.4 | 0.60±0.01 | 4.086±0.172 | 4.360-4.846 |

Table 2. The $^{14}$C dating of 135.8 m Chongce ice core. Absolute uncertainties are given as 1σ range.

| Sample # | Depth (m) | Mass (g) | WIOC (μg) | F$^{14}$C | $^{14}$C age (ka B.P.) | Cal. age (ka B.P.) |
|---|---|---|---|---|---|---|
| CB-1 | 79.46-80.21 | 307.7 | 20.3±1.2 | 0.81±0.01 | 1.679±0.078 | 1.445-1.704 |
| CB-2 | 88.82-89.56 | 302.9 | 24.3±1.4 | 0.80±0.01 | 1.831±0.138 | 1.572-1.921 |
| CB-3 | 99.44-100.10 | 304.6 | 13.8±0.9 | 0.68±0.01 | 3.133±0.161 | 3.157-3.560 |
| CB-4 | 110.58-111.35 | 342.6 | 24.9±1.4 | 0.78±0.01 | 2.037±0.142 | 1.827-2.296 |
| CB-5 | 116.62-117.43 | 330.9 | 9.1±0.7 | 0.69±0.01 | 3.012±0.164 | 2.978-3.377 |
| CB-6 | 122.64-123.36 | 338.6 | 17.6±1.1 | 0.69±0.01 | 2.944±0.157 | 2.892-3.331 |
| CB-7 | 131.41-132.10 | 324.6 | 22.6±1.3 | 0.59±0.01 | 4.228±0.176 | 4.451-5.036 |
| CB-8 | 132.65-133.51 | 392.7 | 23.6±1.4 | 0.60±0.01 | 4.169±0.175 | 4.424-4.951 |
| CB-9 | 134.31-135.03 | 292.4 | 23.0±1.4 | 0.51±0.01 | 5.466±0.201 | 5.997-6.443 |

For $^3$H and $^{210}$Pb methods, provide uncertainties and references.

Response: $^3$H uncertainties will be included in Figure 2b. The detection limit for $^3$H measurements is <10 TU.

$^{210}$Pb uncertainties are shown in Figure 3 and the related reference is included as:

"Following the widely applied approach described by Gäggeler et al. (1983), the ice age was derived using the constant initial concentration (CIC) model."

5) results About $\beta$-activity profile, how are you sure about the date of each peak without additional information? Did you use annual layer counting to theses depth? What do you know about surface age, did any annual layers disappeared?

Response: We did have annual layer counting from stable isotopes to back up the *β*-activity horizons. For more details, please see our previous paper (An et al., 2016).

As to the surface age, we have examined the area changes of glaciers on the Western Kunlun Mountain (including the Chongce ice cap) since the 1970s (Wang, Y., Hou, S., Huai, B., An, W., Pang, H., Liu, Y.: Glacier anomaly over the Western Kunlun Mountains, northwestern Tibetan Plateau, since the 1970s, J. Glaciol., 3[rd] revision). For the whole area, change of the glacier area reveals insignificant shrinkage by $0.07 \pm 0.1\%$ yr$^{-1}$ from the 1970s to 2016. Although the Chongce glacier retreated between 1977 and 1990, and advanced from 1990 to 2011, then remained stable until 2016, the Chongce ice cap remained static from the 1977 to 2016 (Fig. 4), confirming the stability of the ice cap where our ice cores were recovered. The estimate of elevation changes over the West Kunlun Mountain by Lin et al. (2017) and Zhou et al. (2018) (Figs 5 and 6) is also roughly in balance between 1973 and 2014. Therefore, we regard the surface age as the drilling time, and no annual layer disappeared. Moreover, the ice core density profiles (Fig. 2) and the borehole temperature profiles (Fig. 3) also suggest a sounding condition for preserving the ice core records.

[Figure]

Fig. 4. Map showing the Chongce Ice Cap (CIC) and the Chongce Glacier (CG), with the terminus positions at different time. The star shows the position of the drilling site of the Chongce Cores 2 and 3, which might be an optimal site for retrieving an undisturbed paleoclimate record. The inset is from Fig. 3 of Yasuda and Furuya (2015) with the red area showing the surged area confined within the Chongce glacier.

[Figure]

Fig.5. Glacier height changes from 2000 to the 2010s from Lin et al. (2017)

[Figure]

Fig.6. Glacier elevation change from 1973 to 2010 from Zhou et al. (2018)

For $^3$H profile in figure 2b, it seems for me that the plotted values are corrected from decay to reach such high numbers? 3237TU for the 1963 peak value is very high in comparison with other ice core data, mostly around few hundreds TU today value for the 1963 peak, see Gelaidaindong core (Kang, 2015). Give reference on the identification of maximum peak.

Response: Yes the values are corrected for the decay since 1963 as indicated on the y-axis in manuscript Fig. 2b. The depth of the sample with the highest activity was related to the year 1963, the year that the atmospheric test ban treaty was signed and tritium levels in precipitation began to decline gradually because of radioactive decay and the cessation of atmospheric testing (e.g. Kendall and Doctor, 2003). We will make a revision accordingly.

$^{14}$C data interpretation: can you argue why in C4, 14C dates are only around 1-2ka for the intermediate depth (80-170m) and between 2-4ka in C2 at 60-110m? For me C4 is discontinuous, may be because of the particular bedrock topography (presence of bedrock canyon), with recent ice on the 0-110m over older dead ice.

Response: $^{14}$C measurements for samples with relatively young ages are challenging and limited by the currently achieved analytical precision using WIOC in combination with the relatively flat $^{14}$C calibration curve, especially for samples younger than ~1 ka. For more details addressing this comment please see response to comment 1) above. In any case, we agree that additional independent evidences (e.g., stable isotopes, methane) would be beneficial to decipher the $^{14}$C results obtained for the intermediate depth, which will be presented in a separate manuscript.

The use of a simplistic 2-p model is not appropriate for this kind of site where you know that the accumulation is not constant through the Holocene, but especially to be used on the bottom of the glacier where in principle its exponential approach is unrealistic. You have enough [14]C data to trigger glacier bottom age using a simple exponential regression down to bedrock depth for C4.

Response: We agree that the applied 2-p model is limited, especially for the bottom of the glacier where it becomes infinite as described in the manuscript. As discussed above (see response to point 1), we used the flow model by constraining it to fit the dating points in order to obtain a continuous age-depth scale. To avoid overfitting of the data and giving to much weight to individual data points, we prefer not to make assumptions about changes in accumulation, such as by applying a Monte-Carlo approach (Uglietti et al., 2016; Gabrielli et al., 2016). We are confident that our careful and rather conservative approach results in a robust estimate with sufficient precision for the general discussion and conclusions. Nevertheless, we are aware that the approach of extrapolating the age scale from the oldest [14]C sample down to bedrock is far from satisfatory, a fact clearly stated in the manuscript. Therefore, this part will be moved to the supplement.

If applying a simple exponential regression, the simulated ages at bedrock are 3.987 ka B.P. and 5.395 ka ka B.P. for Core 4 and Core 2, respectively, younger than their respective oldest cal. [14]C ages (Figs 7a and 8a), which may be biased by the younger [14]C measurements. Taking this into consideration, we performed more runs of exponential regression by subsequently deleting the uppermost [14]C measurement in each consecutive run. So the last run was based on the lowest cal. [14]C age and the ages from the tritium horizon and [210]Pb for Core

4). For Core 2, the last run was based on the lowest cal. $^{14}$C age and the age from $\beta$-activity horizon, as well as the surface age in order to fix the regression. The simulated ages at bedrock for each run are presented in Tables 3 and 4, giving age ranges of 3.99-5.59 ka B. P. for Core 4 and 5.40-7.67 ka B. P. for Core 2, respectively. These results, although at the lower end, are well within the 1 sigma range of our results presented in the initial manuscript and based on the approach discussed therein.

[Figure]

Fig. 7. Profiles of an exponential regression for Core 4, (a) including all the cal. $^{14}$C ages, (b) including the four lowest cal. $^{14}$C ages to result in the oldest age at the bottom among all the runs (Table 3). The dashed lines represent the 1σ confidence interval of the regression (solid line). The red cross stands for the tritium horizon, green diamonds for the $^{210}$Pb ages calculated at intervals of 5 m w.e., and the blue dots for the cal. $^{14}$C ages with 1σ error bar.

[Figure]

Fig. 8. Profile of an exponential regression for Core 2, (a) including all the cal. $^{14}$C ages, (b) including the two lowest cal. $^{14}$C ages to result in the oldest age at the bottom among all the runs (Table 4). The dashed lines represent the 1σ confidence interval of the regression (solid line). The red cross stands for the *β*-activity horizon and the blue dots for the cal. $^{14}$C ages with 1σ error bars.

Table 3. The simulated ages at bedrock by applying an exponential regression ($y=ae^{bx}$) for Core 4.

| No. of cal. $^{14}$C ages used | a | b | $R^2$ | Simulated ages at bedrock (ka B. P.) |
|---|---|---|---|---|
| 22 | 0.00999 | 0.03327 | 0.79756 | 3.99 |
| 21 | 0.00956 | 0.03352 | 0.79466 | 3.99 |
| 20 | 0.00902 | 0.03386 | 0.79182 | 4.00 |
| 19 | 0.00802 | 0.03453 | 0.79121 | 4.02 |
| 18 | 0.00714 | 0.03520 | 0.78944 | 4.04 |
| 17 | 0.00507 | 0.03716 | 0.79961 | 4.08 |
| 16 | 0.00371 | 0.03895 | 0.80400 | 4.13 |
| 15 | 0.00212 | 0.04214 | 0.82176 | 4.20 |
| 14 | $818311 \times 10^{-9}$ | 0.04754 | 0.86176 | 4.30 |
| 13 | $525863 \times 10^{-9}$ | 0.05004 | 0.86651 | 4.34 |
| 12 | $313385 \times 10^{-9}$ | 0.05298 | 0.87189 | 4.39 |
| 11 | $975051 \times 10^{-10}$ | 0.05958 | 0.89357 | 4.50 |
| 10 | $581575 \times 10^{-10}$ | 0.06249 | 0.89367 | 4.54 |
| 9 | $581236 \times 10^{-11}$ | 0.07547 | 0.92826 | 4.72 |
| 8 | $104984 \times 10^{-11}$ | 0.08509 | 0.94233 | 4.85 |
| 7 | $588197 \times 10^{-13}$ | 0.10128 | 0.97316 | 5.05 |
| 6 | $554846 \times 10^{-14}$ | 0.11453 | 0.98125 | 5.20 |
| 5 | $266867 \times 10^{-15}$ | 0.13153 | 0.98595 | 5.39 |
| 4 | $694579 \times 10^{-17}$ | 0.15195 | 0.98657 | 5.59 |
| 3 | 0.03597 | 0.02704 | 0.99826 | 4.67 |
| 2 | 0.04418 | 0.02600 | 0.99968 | 4.76 |
| 1 | 0.04255 | 0.02614 | 0.99987 | 4.70 |

Table 4. The simulated ages at bedrock by applying an exponential regression ($y=ae^{bx}$) for Core 2.

| No. of cal. $^{14}$C ages used | a | b | $R^2$ | Simulated ages at bedrock (ka B. P.) |
|---|---|---|---|---|
| 9 | 0.19181 | 0.02983 | 0.84957 | 5.40 |
| 8 | 0.16245 | 0.03139 | 0.83787 | 5.45 |
| 7 | 0.13711 | 0.03297 | 0.82107 | 5.49 |
| 6 | 0.02219 | 0.04981 | 0.94779 | 5.88 |
| 5 | 0.02555 | 0.04852 | 0.94051 | 5.86 |
| 4 | 0.00269 | 0.06904 | 0.96654 | 6.18 |
| 3 | $922368 \times 10^{-11}$ | 0.12049 | 0.97807 | 6.83 |
| 2 | $414342 \times 10^{-15}$ | 0.21070 | 0.99977 | 7.67 |
| 1 | 0.00247 | 0.07051 | 1 | 6.70 |

Lines 234 and 245, you indicate tritium when it is from $\beta$-activity data.

Response: We apologize for this negligence. Correction will be made accordingly.

Dating of C2: you are absolutely trying to match a curve from an unsuitable model with data from real measurements, and in addition you add values from other sites to try to force your conclusions. This reasoning and this method are not acceptable. Given the weakness of the model, it is reasonable to remove it from this manuscript.

Response: Although we understand the concern of the reviewer regarding the use of additional constraint for Core 2, our main point is the aim to get the most robust age estimates based on the available data without overfitting or overweighting individual dating points

considering all the potential uncertainties. Since a more advanced glacier ice flow model (i.e.

3D) model is not available and in fact would face similar uncertainties close to the bedrock

(e.g. Luthi and Funk, 2000), what we did is to carefully evaluate our age model output by

taking advantage of all available data, rather than trying to derive results in order to force our

conclusions. For Core 2, the model is clearly under constrained due to too few dating points.

This result in giving a questionable weight to one single data point (3.3±0.2 ka at ~100 m

depth) which, for currently unknown reasons, might simply be an analytical outlier or be

biased (also see response to comments 1 and 5). Partly because of this reason, the model is

fitted poorly to the oldest/deepest dating point (in manuscript line 235 "the derived age at the

depth of the oldest $^{14}$C sample is $9.1 \pm {}^{7.2}_{4.0}$ ka B.P., much older than the actual $^{14}$C age

(6.3±0.2ka B.P.) at that depth."). By adding additional constraint based on the results from

Core 4, this misfit decreases but we fully agree with the reviewer in the sense that this part of

the approach is far from satisfactory. As discussed above, we prefer to move the extrapolation

down to bedrock into the Supplement.

6) Discussion This chapter is to be reviewed in its entirety. The author must clearly state the

objectives of this manuscript. If it comes to presenting new results from the Chongse Core, in

this case with major corrections these results could be published. If it's a matter of depressing

previous works like those on Guliya glacier, as this manuscript does not bring any innovative

data to this site, not the least substantiated discussion, there is no reason to publish it in TC.

On that stage of the review, various technical corrections can wait for a revised version of the

manuscript.

Response: We will revise the manuscript according to the aforementioned discussion. We will, in a neutral way, present the results of the Chongce ice cores and from other Tibetan ice cores based on their original literature. We will remind the remarkable age range difference of the Guliya and other Tibetan ice cores, and conclude that "more effort is necessary to explore multiple dating techniques to confirm the ages of the TP glaciers, including those from Chongce and Guliya" (Thompson, 2018a).

References

An, W., Hou, S., Zhang, W., Wu, S., Xu, H., Pang, H., Wang, Y., and Liu, Y.: Possible recent warming hiatus on the northwestern Tibetan Plateau derived from ice core records, Sci. Rep., 6, 32813, doi:10.1038/srep32813, 2016.

Fisher, D., Koerner, R., Paterson, W., Dansgaard, W., Gundestrup, N., and Reeh, N.: Effect of wind scouring on climatic records from ice-core oxygen-isotope profiles, Nature, 301, 205-209, doi: 10.1038/301205a0, 1983.

Gabrielli, P. et al.: Age of the Mt. Ortles ice cores, the Tyrolean Iceman and glaciation of the highest summit of South Tyrol since the Northern Hemisphere Climatic Optimum, The Cryosphere, 10, 2779–2797, doi:10.5194/tc-10-2779-2016, 2016.

Gäggeler, H., von Gunten, H. R., Rössler, E., Oeschger, H., and Schotterer, U.: [210]Pb-dating of cold alpine firn/ice cores from Colle Gnifetti, Switzerland, J. Glaciol., 29, 165-177, 1983.

Kang, S., Wang, F., Morgenstern, U., Zhang, Y., Grigholm, B., Kaspari, S., Schwikowski, M., Ren, J., Yao, T., Qin, D., and Mayewski, P.: Dramatic loss of glacier accumulation

area on the Tibetan Plateau revealed by ice core tritium and mercury records, The Cryosphere, 9, 1213–1222, doi:10.5194/tc-9-1213-2015, 2015.

Kendall, C. and Doctor, D. H.: Stable isotope applications in hydrologic studies, Treatise on Geochemistry, 5, 319-364, doi:10.1016/B0-08-043751-6/05081-7, 2003.

Lin, H., Li, G., Cuo, L., Hooper, A., and Ye, Q.: A decreasing glacier mass balance gradient from the edge of the Upper Tarim Basin to the Karakoram during 2000–2014, Sci. Rep., 7, 612, doi:10.1038/s41598-017-07133-8, 2017.

Luthi, M. and Funk, M.: Dating ice cores from a high Alpine glacier with a flow model for cold firn, Ann. Glaciol., 31, 69-79, doi:10.3189/172756400781820381, 2000.

Raynaud, D., Barnola, J., Chappellaz, J., Blunier, T., Indermühle, A., and Stauffer, B.: The ice core record of greenhouse gases: a view in the context of future changes, Quat. Sci. Rev., 19, 9–17, 2000.

Takeuchi, N., Fujita, K., Aizen, V., Narama, C., Yokoyama, Y., Okamoto, S., Naoki, K., and Kubota, J.: The disappearance of glaciers in the Tien Shan Mountains in Central Asia at the end of Pleistocene, Quat. Sci. Rev., 103, 26-33, doi:10.1016/j.quascirev.2014.09.006, 2014.

Thompson, L. G., Mosley-Thompson, E., Davis, M., Bolzan, J., Dai, J., Klein, L., Yao, T., Wu, X., Xie, Z., and Gundestrup, N.: Holocene-late pleistocene climatic ice core records from Qinghai-Tibetan Plateau, Science, 246, 474-477, doi:10.1126/science.246.4929.474, 1989.

Thompson, L. G., Yao, T., Davis, M. E., Henderson, K. A., Mosley-Thompson, E., Lin, P.-N., Beer, J., Synal, H.-A., Cole-Dai, J., and Bolzan, J.F.: Tropical climate instability: the

last glacial cycle from a Qinghai-Tibetan ice core, Science, 276, 1821-1825, doi: 10.1126/science.276.5320.1821, 1997.

Thompson, L., Yao, T., Mosley-Thompson, E., Davis, M., Henderson, K., Lin, P.: A high-resolution millennial record of the south Asian monsoon from Himalayan ice cores. Science 289, 1916-1919, 2000.

Thompson, L. G., Davis, M., Mosley-Thompson, E., Lin, P., Henderson, K., and Mashiotta, T.: Tropical ice core records: evidence for asynchronous glaciation on Milankovitch timescales, J. Quat. Sci., 20, 723-733, 2005.

Thompson, L. G., Yao, T., Davis, M., Mosley-Thompson, E., Mashiotta, T., Lin, P., Mikhalenko, V., and Zagorodnov, V.: Holocene climate variability archived in the Puruogangri ice cap on the central Tibetan Plateau, Ann. Glaciol., 43, 61-69, 2006.

Thompson, L.: Interactive comment on "Age of the Tibetan ice cores" by Shugui Hou et al., The Cryosphere Discuss., doi:10.5194/tc-2018-55-SC1, 2018a.

Thompson, L., Yao, T., Davis, M., Mosley-Thompson, E., Wu, G., Porter, S., Xu, B., Lin, P., Wang, N., Beaudon, E., Duan, K., Sierra-Hernández, M., and Kenny, D.: Ice core records of climate variability on the Third Pole with emphasis on the Guliya ice cap, western Kunlun Mountains, Quat. Sci. Rev., 188, 1-14, doi:10.1016/j.quascirev.2018.03.003, 2018b.

Uglietti, C., Zapf, A., Jenk, T., Sigl, M., Szidat, S., Salazar, G., and Schwikowski, M.: Radiocarbon dating of glacier ice: overview, optimisation, validation and potential, Cryosphere 10, 3091-3105, doi:10.5194/tc-10-3091-2016, 2016.

Yao, T., Duan, K., Xu, B., Wang, N., and Pu, J.: Temperature and methane changes over the past 1000 years recorded in Dasuopu glacier (central Himalaya) ice core, Ann. Glaciol., 35, 379–383, 2002.

Yao, T., Masson-Delmotte, V., Gao, J., Yu, W., Yang, X., Risi, C., Sturm, C., Werner, M., Zhao, H., He, Y., Ren, W., Tian, L., Shi, C., and Hou, S.: A review of climatic controls on $\delta 18O$ in precipitation over the Tibetan Plateau: Observations and simulations, Rev. Geophys., 51, 525–548, doi:10.1002/rog.20023, 2013.

Yasuda, T. and Furuya, M.: Dynamics of surge-type glaciers in West Kunlun Shan, Northwestern Tibet, J. Geophys. Res. Earth Surf., 120, 2393–2405, doi: 10.1002/2015JF003511, 2015.

Zhou, Y., Li, Z., Li, J., Zhao, R., and Ding, X.: Glacier mass balance in the Qinghai–Tibet Plateau and its surroundings from the mid-1970s to 2000 based on Hexagon KH-9 and SRTM DEMs, Remote Sens. Environ., 210, 96-112, doi: 10.1016/j.rse.2018.03.020, 2018.

---

## Referee Comment (RC2) · P. A. Mayewski (Referee) · 25 May 2018

1. Does the paper address relevant scientific questions within the scope of TC? – Yes, it calls into question conclusions based on the interpretation of a single water soluble 36Cl measurement that has been used to provide a reference point in following publications to assert timing and extent of past glacial age ice cover and associated climate interpretations. 2. Does the paper present novel concepts, ideas, tools, or data? Yes it provides an impressive rationale and data assembly for dating a relatively new Tibetan ice core record – Chongce. 3. Are substantial conclusions reached? Yes, the authors present a clearand straightforward comparison of existing Tibetan ice core records. 4. Are the scientific methods and assumptions valid and clearly outlined? Yes. 5. Are the results sufficient to support the interpretations and conclusions? The interpretations and conclusions are based on significant spatial and temporal data and the basic conclusion is very professionally stated calling for more work, ideally similar to that

presented in this paper, re dating of the Guliya ice core. 6. Is the description of experiments and calculations sufficiently complete and precise to allow their reproduction by fellow scientists (traceability of results)? Yes 7. Do the authors give proper credit to related work and clearly indicate their own new/original contribution? Yes 8. Does the title clearly reflect the contents of the paper? Yes 9. Does the abstract provide a concise and complete summary? Yes 10. Is the overall presentation well structured and clear? Absolutely. 11. Is the language fluent and precise? Yes 12. Are mathematical formulae, symbols, abbreviations, and units correctly defined and used? Yes 13. Should any parts of the paper (text, formulae, figures, tables) be clarified, reduced, combined, or eliminated? No. 14. Are the number and quality of references appropriate? Yes. 15. Is the amount and quality of supplementary material appropriate? Yes.

---

## Author Comment (AC3) · 27 May 2018

Dear Prof. Paul Mayewski,

Many thanks for the review. We appreciate.

Best regards,

Hou